



Nonlinear Processes
in Geophysics

# A methodology to obtain model-error covariances due to the discretization scheme from the parametric Kalman filter perspective

**Olivier Pannekoucke**[1,2,3], **Richard Ménard** [TS1][4], **Mohammad El Aabaribaoune**[2,3], **and Matthieu Plu**[2]

[1]INPT-ENM, Toulouse, France
[2]CNRM, Université de Toulouse, Météo-France, CNRS, Toulouse, France
[3]CERFACS, Toulouse, France
[4]ARQI/Air Quality Research Division, Environment and Climate Change Canada, Dorval, Québec, Canada

**Correspondence:** Olivier Pannekoucke (olivier.pannekoucke@meteo.fr)

**Abstract.** This contribution addresses the characterization of the model-error covariance matrix from the new theoretical perspective provided by the parametric Kalman filter method which approximates the covariance dynamics from the parametric evolution of a covariance model. The classical approach to obtain the modified equation of a dynamics is revisited to formulate a parametric modelling of the model-error covariance matrix which applies when the numerical model is dissipative compared with the true dynamics. As an illustration, the particular case of the advection equation is considered as a simple test bed. After the theoretical derivation of the predictability-error covariance matrices of both the nature and the numerical model, a numerical simulation is proposed which illustrates the properties of the resulting model-error covariance matrix.

## 1 Introduction

A significant portion of the work being carried out in state-of-the-art data assimilation concerns the treatment of the forecast-error covariance matrix. Actually, the forecast error is composed of two parts. While one part of it is related to the uncertainty in the initial condition, another part is due to the model error (Daley, 1991; Dee, 1995). The model error corresponds to the difference between the simulation and the true behaviour of a system, and several representations of the model error can be introduced in numerical weather prediction (NWP) (Houtekamer et al., 2009). For instance, the model error can be related to the misrep-

resentation of the small scales and how this influences the large scales. Stochastic physics such as stochastic kinetic energy backscatter (Shutts, 2005) or the stochastically perturbed parametrization tendencies (Palmer et al., 2009) are examples of methods encountered in NWP for this part of the model error.

Although some theoretical studies have been conducted in the past, which elucidate the generic behaviour related to the model error from the dynamical system perspective and in connection with the data assimilation (e.g. Nicolis, 2003; Vannitsem and Toth, 2002; Carrassi and Vannitsem, 2010), as far as we know there has been little investigation of the effect of the discretization of partial derivative equations on the model error and on model-error covariance in particular (Dubinkina, 2018; Hatfield et al., 2018; Grudzien et al., 2020). One reason why the effect of numerical schemes is rarely considered is because it tends to be quite difficult to describe the dynamics of large covariance matrices as encountered in the Kalman filter.

It has been noted in Kalman filtering and ensemble Kalman filtering (EnKF) that the propagation of error covariance with a discretized advection model produces a model error (variance) in the form of a variance loss (Ménard et al., 2000, 2020). This error is related to the spatial splitting error in covariance propagation that exists with discretized models and not in continuous propagation of covariance functions, i.e. the propagation by the true equations of the dynamics.

Recently, Pannekoucke et al. (2016, 2018b) (P16) have proposed to solve the Kalman filter equations, and their second-order extension for non-linear dynamics, using ap-

Please note the remarks at the end of the manuscript.

proximated covariance matrices through a covariance model characterized by certain parameters, leading to the so-called "parametric Kalman filter" (PKF). With this approximation, the dynamics of the covariances is replaced by the dynamics of the parameters. For instance, when considering the class of covariance matrices parameterized by the variance field and the local anisotropic tensors (VLATcov), the evolution of the matrices is deduced from the evolution of the variance and the local anisotropic tensors (Cohn, 1993; Pannekoucke, 2020). This approach relies on the partial differential equations encountered in geosciences that are often non-linear.

The aims of the present work are to study how the parametric dynamics for covariance matrix evolution can help to characterize the model-error covariance matrix, and more precisely, to determine if is it possible to capture some part of the model-error covariance which is due to the numerical scheme. In this methodological contribution, we will limit ourselves to diffusive numerical errors whose uncertainty dynamics can be explored from the results of Pannekoucke et al. (2018a) (P18).

The paper is organized as follows: the background in data assimilation is reviewed in Sect. 2 from which the formalism of the model-error covariance matrix is detailed with the introduction of modelling that could apply when the numerical model is dissipative. The model-error covariance matrix based on the PKF is illustrated for the particular one-dimensional transport equation in Sect. 3 in the context of the Euler-upwind and semi-Lagrangian schemes. A numerical test bed is proposed in Sect. 4 to assess the ability of the PKF approach to successfully model the flow-dependent part of the model-error covariance matrix to numerical schemes in a one-dimensional setting. A discussion on the results is proposed in Sect. 5. Conclusions and perspectives are given in Sect. 6.

## 2   Theoretical considerations

### 2.1   Background in uncertainty propagation and the model error

Here, we assume that the *nature* is governed by the deterministic equation

$$\partial_t \mathcal{X} = \mathcal{N}(\mathcal{X}),    \tag{1}$$

where $\mathcal{X}$ stands for the state. Note that $\mathcal{X}$ can be either discrete or continuous: the discrete case leads to matrix of algebraic relations, while the continuous case is suitable for the oretical treatment with partial differential equations. Thereafter, for any state $\mathcal{X}$ of a suitable set, there exists a single trajectory $\mathcal{X}_t = \mathcal{N}_{t \leftarrow 0}(\mathcal{X})$, where $\mathcal{N}_{t \leftarrow 0}$ stands for the propagator of the dynamics (Eq. 1) from 0 to $t$. Hence, if $\mathcal{X}_q^t$ denotes the true state of the nature at time $t_q$, then the true state of the nature at time $t_{q+1}$ is

$$\mathcal{X}_{q+1}^t = \mathcal{N}_{t_{q+1} \leftarrow t_q}(\mathcal{X}_q^t),    \tag{2}$$

where the subscript $q$ is used to denote the time $t_q$.

Due to the imperfect knowledge of the nature and the limitations encountered during the computation, the nature dynamics is only approximated by

$$\partial_t \mathcal{X} = \mathcal{M}(t, \mathcal{X}),    \tag{3}$$

where $\mathcal{M}$ is the numerical dynamics. Compared with the nature, the time evolution of the true state (Eq. 2) is now related to the numerical dynamics as

$$\mathcal{X}_{q+1}^t = \mathcal{M}_{t_{q+1} \leftarrow t_q}(\mathcal{X}_q^t) - \varepsilon_{q+1}^m(\mathcal{X}_q^t),    \tag{4}$$

where $\varepsilon_{q+1}^m(\mathcal{X}_q^t)$ is the *model error* with respect to the true state, and where $\varepsilon_{q+1}^m$ is the vector-valued function defined by

$$\varepsilon_{q+1}^m = \mathcal{M}_{t_{q+1} \leftarrow t_q} - \mathcal{N}_{t_{q+1} \leftarrow t_q}.    \tag{5}$$

The model error $\varepsilon_{q+1}^m(\mathcal{X}_q^t)$ represents collectively the numerical discretization error and the effect of unresolved processes. It is often modelled as a random field of zero mean, i.e. $\mathbb{E}\left[\varepsilon_{q+1}^m(\mathcal{X}_q^t)\right] = 0$, and of covariance matrix

$$\mathbf{P}_{q+1}^m = \mathbb{E}\left[\varepsilon_{q+1}^m(\mathcal{X}_q^t)\left(\varepsilon_{q+1}^m(\mathcal{X}_q^t)\right)^T\right],    \tag{6}$$

where $\mathbb{E}[\cdot]$ denotes the expectation operator.

In practice, the true state $\mathcal{X}_q^t$ is unknown and only an estimation can be deduced from a priori information and the available observations. This estimation is called the "analysis state", $\mathcal{X}^a$, and it is expanded as

$$\mathcal{X}_q^a = \mathcal{X}_q^t + \varepsilon_q^a,    \tag{7}$$

where $\varepsilon_q^a$ stands for the so-called "analysis error" that is modelled as a random field of zero mean and covariance matrix $\mathbf{P}^a = \mathbb{E}\left[\varepsilon_q^a(\varepsilon_q^a)^T\right]$. The forecast state is the prediction made from the analysis state,

$$\mathcal{X}_{q+1}^f = \mathcal{M}_{t_{q+1} \leftarrow t_q}(\mathcal{X}_q^a).    \tag{8}$$

Similarly to the analysis state, the forecast state expands as

$$\mathcal{X}_{q+1}^f = \mathcal{X}_{q+1}^t + \varepsilon_{q+1}^f,    \tag{9}$$

where $\varepsilon_{q+1}^f$ stands for the so-called "forecast error" that is modelled as a random field of zero mean and covariance matrix $\mathbf{P}^f = \mathbb{E}\left[\varepsilon_{q+1}^f(\varepsilon_{q+1}^f)^T\right]$.

The forecast-error covariance matrix is related to the analysis-error covariance matrix through a deterministic relation as follows. From the definition of the forecast error (Eq. 9), its dynamics is given by (see Eq. A2 in Appendix A)

$$\varepsilon_{q+1}^f = \mathbf{M}\varepsilon_q^a + \varepsilon_{q+1}^m(\mathcal{X}_q^t),    \tag{10}$$

where $\mathbf{M}$ is a simplified notation for $\mathbf{M}_{t_{q+1} \leftarrow t_q, \mathcal{X}_q^a}$ that is for the tangent linear (TL) propagator along the analysis trajectory. The TL dynamics, with respect to the analysis state, is defined by

$$\partial_t \varepsilon = \mathbf{M}_{t, \mathcal{X}_t^a} \varepsilon, \tag{11}$$

where $\mathbf{M}_{t, \mathcal{X}_t^a} = d\mathcal{M}_{|t, \mathcal{X}_t^a}$ is the differential of $\mathcal{M}$ at $(t, \mathcal{X}_t^a)$. This TL model governs the evolution of small perturbations along the forecast trajectory starting from the analysis state. Note that the validity of the TL dynamics depends on the error magnitude and on the forecast range. Moreover, the decomposition of the forecast error (Eq. 10) makes the predictability error $\varepsilon^p$ TS2 appear defined by

$$\varepsilon_{q+1}^p = \mathbf{M} \varepsilon_q^a. \tag{12}$$

Consequently, the forecast-error covariance matrix becomes

$$\mathbf{P}_{q+1}^f = \mathbf{P}_{q+1}^p + \mathbf{P}_{q+1}^m + \mathbf{V}_{q+1}^{pm} + (\mathbf{V}_{q+1}^{pm})^T, \tag{13}$$

where

$$\mathbf{P}_{q+1}^p = \mathbf{M} \mathbf{P}_q^a \mathbf{M}^T \tag{14}$$

is the predictability-error covariance matrix (Daley, 1992) and $\mathbf{V}_{q+1}^{pm} = \mathbb{E}\left[\varepsilon_{q+1}^p \left(\varepsilon_{q+1}^m\right)^T\right]$ TS3 denotes the cross-covariance matrix between the predictability error and the model error.

When the analysis error and the model error are decorrelated, the forecast-error covariance matrix is written as

$$\mathbf{P}_{q+1}^f = \mathbf{P}_{q+1}^p + \mathbf{P}_{q+1}^m. \tag{15}$$

Note that, in the case where the true nature is used to forecast the uncertainty, the forecast-error covariance matrix coincides with the predictability-error covariance matrix. In the latter, the predictability error with respect to the nature dynamics plays an important role. So in order to avoid any confusion with the predictability error associated with the numerical model, the notation $\widetilde{\cdot}$ is used when the dynamics is the nature, i.e.

$$\widetilde{\mathbf{P}}_{q+1}^f = \widetilde{\mathbf{P}}_{q+1}^p = \mathbf{N} \mathbf{P}_q^a \mathbf{N}^T, \tag{16}$$

with $\widetilde{\mathbf{P}}^f = \mathbb{E}\left[\widetilde{\varepsilon}^f (\widetilde{\varepsilon}^f)^T\right]$, where $\widetilde{\varepsilon}_{q+1}^f = \mathbf{N} \varepsilon_q^a$ denotes the forecast error in the particular case where the dynamics is the nature, which coincide with the predictability error in this case, i.e. $\widetilde{\varepsilon}_{q+1}^f = \widetilde{\varepsilon}_{q+1}^p$; and where $\mathbf{N}$ is a simplified notation for the propagator $\mathbf{N}_{t_{q+1} \leftarrow t_q, \mathcal{X}_q^a}$ solution of the TL dynamics governed by $\mathbf{N}_{t, \mathcal{X}_t^a} = d\mathcal{N}_{|t, \mathcal{X}_t^a}$ (the differential of $\mathcal{N}$ at $(t, \mathcal{X}_t^a)$).

## 2.2 Discussion on the modelling of the model error

The modelling of the model error can be seen as a trade-off between its real properties and the lack of knowledge to address this error. In particular, the various assumptions encountered in data assimilation may be considered as suboptimal ways to model this error. For instance, assuming that the model error is unbiased leads to modelling the bias as some variance and overestimates the effective model-error variance. Then, assuming a decorrelation between the analysis and the model errors is certainly wrong for deterministic error, as for the model error due to the discretization of the dynamics, but it may not apply for highly non-linear processes as for the turbulent processes and transport by the turbulent processes. Again, assuming the decorrelation between the analysis and the model errors leads to overestimating the true effect of the model error with an overestimation of the true forecast-error uncertainty. However, with these assumptions, or actually this modelling, some part of the model-error statistics can be estimated from the data. For instance, with the assumption that the analysis and the model errors are decorrelated, leading to Eq. (15), it is possible to estimate the homogeneous correlation and the stationary part of the climatological model-error covariance (Daley, 1992; Boisserie et al., 2013).

By some aspects, the understanding and the specification of the model-error covariance matrix look like the development of the background-error covariance matrix some decades ago. Indeed, in variational data assimilation, the background-error covariance matrix was a constant matrix, estimated from the climatology (Derber and Bouttier, 1999). Then, the ensemble methods provided an estimation of the predictability error statistics of the day, leading to a flow-dependent background-error covariance matrix (see, e.g. Berre et al., 2007). Nonetheless, the situation of the model-error covariance matrix is different since, up to now, no equations have been known to characterize its properties. It seems that the prospect of estimating the model-error covariance matrix of the day is out of reach.

Because the model error can mean different things, to understand the context in which we are using model error, let us consider the situation sketched in Fig. 1. This figure mimics the evolution of the analysis uncertainty with respect to the nature and the numerical model. The initial Gaussian analysis error is characterized by the analysis state (the black point) and the analysis-error covariance matrix (the black ellipse). Under the TL assumption, the analysis uncertainty evolving by the nature dynamics (blue arrow) is a Gaussian of mean and covariance given, respectively, by the analysis forecasted by the nature (blue point) and by the predictability-error covariance matrix represented by the blue ellipse. Note that in this case, the predictability-error covariance matrix coincides with the forecast-error covariance matrix. Similarly, the analysis evolving by the numerical model (red arrow) is a Gaussian of mean and covariance given, respectively, by the analysis forecasted by the model (red point) and by the predictability-error covariance matrix represented by the red ellipse. The evolution of the true state (pink crosses) is also represented (pink arrow). Figure 1a and

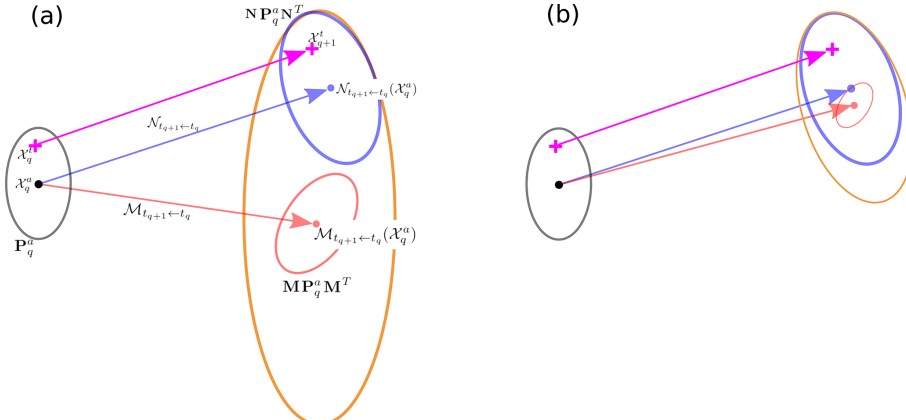

**Figure 1.** Illustration of the evolution of the uncertainty by the nature and the numerical model: the generic situation **(a)** and the particular situation where the forecast lies within the nature uncertainty and where the model is diffusive **(b)**. The predictability-error covariance of the nature $\mathbf{NP}_q^a\mathbf{N}^T$ (of the model $\mathbf{MP}_q^a\mathbf{M}^T$) is indicated by the blue (red) ellipse. The forecast-error covariance matrix is indicated by the orange ellipse.

b illustrate what would be the forecast-error covariance matrix (orange ellipse) in two situations.

Figure 1a represents the case where the forecast state $\mathcal{X}_{q+1}^f$ is out of the predictability uncertainty of the nature: in that case, a model-error estimate is needed to enlarge the predictability-error covariance of the numerical model so that the forecast-error covariance is large enough to account for the uncertainty of the nature. In this situation, it seems difficult to speculate about what would be the characteristic of the model error beyond any climatological estimate. This situation could be the typical picture for a long-term forecast.

Figure 1b represents the situation where the time integration is not too long, so the forecast state lies within the predictability uncertainty of the nature. This situation is encountered when the numerical model is more dissipative than the nature, e.g. the resolution of an advection by a semi-Lagrangian scheme. Then the model-error uncertainty, required to correct the predictability error of the numerical model, should be at least large enough to provide an uncertainty similar to the predictability-error covariance of the nature. So if we are able to quantify the predictability-error covariances of the nature and of the numerical model, then it would be able to specify a flow-dependent part of the model-error covariance matrix. To account for the bias, a climatological residual covariance matrix would be necessary, which corresponds to a static matrix which depends on the duration of the forecast. Note that this decomposition of the model-error covariance matrix into a flow-dependent and a static part should not be confused with the decomposition of the model error itself. In particular, the decomposition of the model-error covariance matrix does not mean that the part of the model error related to the bias is static; this not true, as the bias depends on the situation. However, the estimation of the bias needs the knowledge of the nature dynamics that is never known. Because the statistical contribution of the bias can only be known from a climatological study, this leads to a static matrix which is not flow dependent.

Thereafter, we consider the situation sketched in Fig. 1b, which suggests decomposing the forecast-error covariance matrix as

$$\mathbf{P}_{q+1}^f \approx \mathbf{P}_{q+1}^p + \mathbf{\Pi}_{q+1}^m + \mathbf{Q}_{q+1}, \tag{17}$$

where

$$\mathbf{\Pi}_{q+1}^m = \mathbf{NP}_q^a\mathbf{N}^T - \mathbf{MP}_q^a\mathbf{M}^T \tag{18}$$

would account for the flow-dependent part of the model-error covariance matrix, while the remaining $\mathbf{Q}_{q+1}$, a residual model-error covariance, would account for the bias and could be estimated from the climatology, e.g. by considering a chi-squared diagnostic (Ménard et al., 2000).

Thus, $\mathbf{\Pi}_{q+1}^m = \widetilde{\mathbf{P}}_{q+1}^p - \mathbf{P}_{q+1}^p$ measures how the predictability-error covariance of the numerical model should be modified to find the one of the nature. We think that $\mathbf{\Pi}_{q+1}^m$ could be a useful proxy to characterize the flow-dependent part of model-error covariance matrix. Note that the matrix $\mathbf{\Pi}_{q+1}^m$ is symmetric but not necessarily positive. However, under the assumption depicted in Fig. 1b, we will assume that $\mathbf{\Pi}_{q+1}^m$ is positive. Note also that $\mathbf{\Pi}_{q+1}^m$ is different from the model-error covariance matrix $\mathbf{P}_{q+1}^m$: if there is no analysis uncertainty, then $\mathbf{\Pi}_{q+1}^m$ is zero.

The decomposition (Eq. 17) can be justified from the decomposition of the forecast error that can be written as (see Eq. A6 in Appendix A)

$$\boldsymbol{\varepsilon}_{q+1}^f = \widetilde{\boldsymbol{\varepsilon}}_{q+1}^p + \varepsilon_{q+1}^m(\mathcal{X}_q^a), \tag{19}$$

which makes the forecast error appear, $\boldsymbol{\varepsilon}_{q+1}^f$, as the predictability error of the nature, $\widetilde{\boldsymbol{\varepsilon}}_{q+1}^p = \mathbf{N}\boldsymbol{\varepsilon}_q^a$, plus a drift $\varepsilon_{q+1}^m(\mathcal{X}_q^a)$. Note that, with the analysis state $\mathcal{X}_q^a$ being known,

the model error $\varepsilon_{q+1}^{\mathrm{m}}(\mathcal{X}_q^{\mathrm{a}})$ is easier to handle than $\varepsilon_{q+1}^{\mathrm{m}}(\mathcal{X}_q^{\mathrm{t}})$ in Eq. (10), which is defined with respect to the true state $\mathcal{X}_q^{\mathrm{t}}$ that is never known in practice. Now, when assuming that the errors in Eq. (19) are decorrelated and when the model error $\varepsilon_{q+1}^{\mathrm{m}}(\mathcal{X}_q^{\mathrm{a}})$ is unbiased ($\mathbb{E}\left[\varepsilon_{q+1}^{\mathrm{m}}(\mathcal{X}_q^{\mathrm{a}})\right] = 0$) and of covariance $\mathbb{E}\left[\varepsilon_{q+1}^{\mathrm{m}}(\mathcal{X}_q^{\mathrm{a}})\left(\varepsilon_{q+1}^{\mathrm{m}}(\mathcal{X}_q^{\mathrm{a}})\right)^T\right] = \mathbf{Q}_{q+1}$, it results that the forecast-error covariance matrix is also written as

$$\mathbf{P}_{q+1}^{\mathrm{f}} = \widetilde{\mathbf{P}}_{q+1}^{\mathrm{p}} + \mathbf{Q}_{q+1}. \tag{20}$$

Hence, the modelling of the model-error covariance as

$$\mathbf{P}_{q+1}^{\mathrm{m}} \approx \mathbf{\Pi}_{q+1}^{\mathrm{m}} + \mathbf{Q}_{q+1} \tag{21}$$

allows us to connect the two formulations (Eqs. 15 and 20) of $\mathbf{P}_{q+1}^{\mathrm{f}}$. In fact, while Eqs. (15) and (20) result from a decorrelation assumption of the errors in Eqs. (10) and (19), and because $\mathbf{\Pi}_{q+1}^{\mathrm{m}}$ is not necessarily a covariance matrix, then expression of $\mathbf{P}_{q+1}^{\mathrm{f}}$, proposed in Eq. (17), is more like that of Eq. (13), where there is no decorrelation assumption.

Compared with climatological modelling of the model-error covariance matrix, as usually encountered in data assimilation, the model for $\mathbf{P}^{\mathrm{m}}$ in Eq. (21) is a state-dependent model of the model-error covariance. Note also that, in Fig. 1b, assuming that there is no bias, while there is one, leads to interpret the bias as a residual model-error whose magnitude can be estimated from the climatology. Hence, $\mathbf{P}^{\mathrm{m}}$ modelled by Eq. (21) is a hybrid model that balances the model error of the day with the climatological effects of the model error. In particular, if the initial state is perfectly known, then $\mathbf{\Pi}_{q+1}^{\mathrm{m}}$ is zero, and the model error is characterized by the climatological residual term $\mathbf{Q}_{q+1}$: the source of this uncertainty corresponds to a forcing term that appears in the dynamics of the model error (see, e.g. Eq. 4 in Nicolis, 2003); this source term is not explored here and its contribution is incorporated in $\mathbf{Q}$ whose magnitude depends on the forecast time.

Note that the modelling equation (Eq. 21) for $\mathbf{P}^{\mathrm{m}}$ is actually supported by at least one real experiment. In the assimilation of a chemical tracer using a Kalman filter, Ménard et al. (2000) and Ménard and Chang (2000) (M2000s) have observed a loss of variance: the variance they forecasted was lower than the theoretical variance that was transported by the flow for the advection equation (Cohn, 1993). Said differently, in their experiment, the predicability-error variance computed from the numerical model was lower than the predicability-error variance of the nature they considered, and M2000s related the loss of variance to the discretization of the continuous dynamics. This loss of variance is also encountered when considering an ensemble forecast of the uncertainty, as later illustrated in the numerical part (see Sect. 4.2.1) and also observed in 3-D domain simulations (Ménard et al., 2020). Accompanying the loss of variance, M2000s also observed that the correlation length scale

they predict was larger, due to the same diffusive process that gives rise to the loss of variance. To cope with the loss of variance, M2000s proposed to correct the predictability-error variance (the diagonal of $\mathbf{P}^{\mathrm{p}}$ in Eq. 14) so that its magnitude is conserved, as it is supposed to be according to the theory. This renormalization introduced an increase of correlation length that was corrected by a Schur product of the new covariances with a homogeneous isotropic correlation model whose length scale has been determined so that the total covariance is conserved over time.

Indeed, M2000s introduced a modelling of the model-error covariance matrix similar to Eq. (21) introduced here, although they did not explicitly formalize it in this way: their objective was not to characterize the model-error covariance matrix but to correct the predictability-error covariance matrix that they considered erroneous from a theoretical point of view.

In particular, M2000s have observed that the Kalman filter, with the corrected predictability-error covariance, required less residual model error $\mathbf{Q}$ (see Ménard et al., 2000, Sect. 5) and improved the analysis-error statistics (see Ménard et al., 2000, Fig. 11): the flow-dependent modelling (Eq. 21) of the model error is in better agreement with the real forecast uncertainty.

At a computational level, $\mathbf{\Pi}_{q+1}^{\mathrm{m}}$ in Eq. (18) appears easier to obtain than the model-error covariance matrix $\mathbf{P}_{q+1}^{\mathrm{m}}$, as defined by Eq. (6): the predictability-error covariance matrices of the model $\mathbf{P}_{q+1}^{\mathrm{p}}$ (Eq. 14) and of the nature $\widetilde{\mathbf{P}}_{q+1}^{\mathrm{p}}$ (Eq. 16) are based only on the TL forecasts with respect to the known analysis state $\mathcal{X}_q^{\mathrm{a}}$, while the model error $\varepsilon_{q+1}^{\mathrm{m}}(\mathcal{X}_q^{\mathrm{t}})$ (Eq. 4) depends on the true state $\mathcal{X}_q^{\mathrm{t}}$ that is never known.

However, computing $\widetilde{\mathbf{P}}_{q+1}^{\mathrm{p}}$ and $\mathbf{P}_{q+1}^{\mathrm{p}}$ remains a challenge. First of all, the nature dynamics $\mathcal{N}$ is generally unknown; e.g. primitive equations are only an approximation of the geophysical fluid dynamics. Then, when the nature dynamics is (assumed) known, e.g. when it is given by partial differential equations (PDEs), there is often no analytical solution, which means that the problem must be solved numerically: as $\mathcal{M}$ is precisely the numerical approximation of $\mathcal{N}$, the only way to compute $\widetilde{\mathbf{P}}_{q+1}^{\mathrm{p}}$ is to introduce a high-order numerical approximation of the nature dynamics, $\widehat{\mathcal{N}}$, whose numerical error is much smaller than the one of $\mathcal{M}$. And finally, it remains to compute $\widetilde{\mathbf{P}}_{q+1}^{\mathrm{p}}$ and $\mathbf{P}_{q+1}^{\mathrm{p}}$. But due to the large size of the numerical state encountered in practice, the direct computation of $\widetilde{\mathbf{P}}_{q+1}^{\mathrm{p}} \approx \widehat{\mathbf{N}}\mathbf{P}_q^a\widehat{\mathbf{N}}^T$ and $\mathbf{P}_{q+1}^{\mathrm{p}} = \mathbf{M}\mathbf{P}_q^a\mathbf{M}^T$ is impossible, even on supercomputers, which are only able to handle a few numerical states at full resolution: it is the limitation that motivated the ensemble estimation to solve the Kalman filter equations (Evensen, 2009).

To overcome the above limitations, a high-order discretization $\widehat{\mathcal{N}}$ of $\mathcal{N}$ will be introduced in the latter numerical simulation in place of $\widetilde{\mathcal{N}}$, e.g. in the ensemble estimation of the covariance matrix $\widetilde{\mathbf{P}}_{q+1}^{\mathrm{p}} \approx \widehat{\mathbf{N}}\mathbf{P}_q^a\widehat{\mathbf{N}}^T$ only used for the validation. But the computation of $\widetilde{\mathbf{P}}_{q+1}^{\mathrm{p}} = \mathbf{N}\mathbf{P}_q^a\mathbf{N}^T$ and $\mathbf{P}_{q+1}^{\mathrm{p}}$ is

investigated through an alternative to the ensemble estimation, as now introduced in the next section.

## 2.3 Parametric dynamics for VLATcov models

The parametric formulation provides a framework where a limited number of covariance parameters (based on the continuous PDE) of the nature can be computed. The parametric formulation works as follows. If $\mathbf{P}(\mathcal{P})$ denotes a covariance model characterized by a set of parameters $\mathcal{P} = (p_i)_{i \in I}$, there exists a set $\mathcal{P}_t^{\mathrm{f}}$ featuring the forecast-error covariance matrix so that $\mathbf{P}(\mathcal{P}_t^{\mathrm{f}}) \approx \mathbf{P}_t^{\mathrm{f}}$; and there is a set $\mathcal{P}^{\mathrm{a}}$ featuring the analysis-error covariance matrix so that $\mathbf{P}(\mathcal{P}^{\mathrm{a}}) \approx \mathbf{P}^a$. In reverse, if the dynamics of the parameters $\mathcal{P}_t^{\mathrm{f}}$ is known, then $\mathbf{P}(\mathcal{P}_t^{\mathrm{f}})$ approximates the dynamics of $\mathbf{P}_t^{\mathrm{f}}$ without using the full matrix computation. This approach constitutes the so-called parametric Kalman filter (PKF) approximation, introduced by Pannekoucke et al. (2016, 2018a) (P16, P18).

The family of covariance models parameterized by the variance field and the local anisotropic tensors, the VLATcov models, are of particular interest (Pannekoucke, 2020): their parameters are directly related to the grid-point statistics of the error field $\varepsilon$. When the error is modelled as an unbiased random differential field, $\mathbb{E}[\varepsilon] = 0$, the variance at a point $\mathbf{x}$ is written as

$$V(\mathbf{x}) = \mathbb{E}\left[\varepsilon(\mathbf{x})^2\right]. \tag{22}$$

The anisotropy of the correlation function $\rho(\mathbf{x}, \mathbf{y}) = \frac{1}{\sqrt{V_{\mathbf{x}} V_{\mathbf{y}}}} \mathbb{E}[\varepsilon(\mathbf{x})\varepsilon(\mathbf{y})]$ is defined, from the second-order expansion,

$$\rho(\mathbf{x}, \mathbf{x} + \delta\mathbf{x}) \approx 1 - \frac{1}{2}||\delta\mathbf{x}||_{\mathbf{g}(\mathbf{x})}^2, \tag{23}$$

by the local metric tensor $\mathbf{g}(\mathbf{x})$. An interesting result is that the metric tensor can be obtained from the error as

$$\mathbf{g}_{ij}(\mathbf{x}) = \mathbb{E}\left[\partial_{x^i}\left(\frac{\varepsilon}{\sqrt{V}}\right)\partial_{x^j}\left(\frac{\varepsilon}{\sqrt{V}}\right)\right] \tag{24}$$

(see, e.g. Pannekoucke, 2020, for details). A VLATcov model is then a covariance model parameterized by $V$ and $\mathbf{g}$, which is $\mathbf{P}(V, \mathbf{g})$.

For instance, the diffusion operator of Weaver and Courtier (2001) is an example of a VLATcov model: the local anisotropic tensors are related to the local diffusion tensors, $\nu$, from

$$\nu_{\mathbf{x}} = \frac{1}{2}\mathbf{g}_{\mathbf{x}}^{-1}, \tag{25}$$

where the superscript $^{-1}$ denotes the matrix inverse operator (Pannekoucke and Massart, 2008; Weaver and Mirouze, 2013). Equation (25) holds under the local homogeneous assumption; that is, when the spatial derivatives are negligible.

Following Pannekoucke et al. (2018a), the parametric dynamics of a VLATcov model is deduced from the dynamics of the errors from

$$\partial_t V = 2\mathbb{E}[\varepsilon \partial_t \varepsilon], \tag{26a}$$

$$\partial_t \mathbf{g}_{ij} = \partial_t \left(\mathbb{E}\left[\partial_{x^i}\left(\frac{\varepsilon}{\sqrt{V}}\right)\partial_{x^j}\left(\frac{\varepsilon}{\sqrt{V}}\right)\right]\right), \tag{26b}$$

where the expectation operator and the temporal derivative commute, $\partial_t \mathbb{E}[\cdot] = \mathbb{E}[\partial_t \cdot]$, as used in Eq. (26a). Therefore, the dynamics of the VLATcov model is written $\mathbf{P}(V_t, \mathbf{g}_t)$ or $\mathbf{P}(V_t, \nu_t)$, which are equivalent.

Now, we apply the parametric covariance dynamics for model-error covariance estimation.

## 2.4 The model-error VLATcov approximation

From now, we will assume that $\mathbf{\Pi}^{\mathrm{m}}$ is a covariance matrix and that there is no residual model-error $\mathbf{Q}$ in order to focus on $\mathbf{\Pi}^{\mathrm{m}}$ alone, so that

$$\mathbf{P}_{q+1}^{\mathrm{m}} \approx \mathbf{\Pi}_{q+1}^{\mathrm{m}} \tag{27}$$

leads to model the forecast-error covariance matrix as

$$\mathbf{P}_{q+1}^{\mathrm{f}} \approx \mathbf{P}_{q+1}^{\mathrm{p}} + \mathbf{\Pi}_{q+1}^{\mathrm{m}}. \tag{28}$$

With the notations of the previous paragraph, a set $\mathcal{P}_t^{\mathrm{p}}$ also exists for the predictability-error covariance matrix, leading to the approximation $\mathbf{P}(\mathcal{P}_t^{\mathrm{p}}) \approx \mathbf{P}_t^p$.

If the dynamics of the parameters $\mathcal{P}_t^{\mathrm{p}}$ is known, then starting from the initial condition $\mathcal{P}_0^{\mathrm{p}} = \mathcal{P}^{\mathrm{a}}$ it is possible to approximately determine $\mathbf{P}_t^p$ without solving Eqs. (14) and (16) explicitly.

Hence, thanks to the parametric dynamics in the case where the nature is known from its partial derivative equations, a new method to compute the model-error covariance matrix can be proposed as follows. By considering the TL dynamics for the numerical model and for the nature, Eq. (26) provides a way to compute both the predictability-error covariance matrices $\mathbf{P}^{\mathrm{p}}$ (Eq. 14) and $\widetilde{\mathbf{P}}^{\mathrm{p}}$ (Eq. 16) from which the model (Eq. 27) of $\mathbf{P}^{\mathrm{m}}$ can be evaluated. For the covariance model based on the diffusion equation, the model-error variance diagnosed from Eq. (18) is the difference:

$$V^{\mathrm{m}} = \widetilde{V}^{\mathrm{p}} - V^{\mathrm{p}}, \tag{29a}$$

where $\widetilde{V}^{\mathrm{p}}$ and $V^{\mathrm{p}}$ denote the predictability-error variance fields of the nature and of the numerical model. The field of the metric tensor of the model error is approximately given by

$$\mathbf{g}^{\mathrm{m}} = \frac{1}{V^{\mathrm{m}}}\left(\widetilde{V}^{\mathrm{p}}\widetilde{\mathbf{g}}^{\mathrm{p}} - V^{\mathrm{p}}\mathbf{g}^{\mathrm{p}}\right), \tag{29b}$$

where $\widetilde{\mathbf{g}}^{\mathrm{p}}$ and $\mathbf{g}^{\mathrm{p}}$, respectively, denote the predictability-error metric tensor fields of the nature and of the numerical model (see Appendix B for details).

In the next section, we apply the parametric model-error dynamics to a transport equation.

## 3 Parametric characterization of the model-error covariance for the one-dimensional advection equation

The transport equation of a passive scalar $c$ by the wind $u(t, x)$ is written as

$$\partial_t c + u \partial_x c = 0, \tag{30}$$

and takes the place of the nature dynamics (Eq. 1). Note that dynamics (Eq. 30) is linear, meaning that the tangent-linear dynamics is also given by Eq. (30). The advection equation has two aspects. The first side is given by the PDE (Eq. 30), which is referred to as the Euler point of view. The other side is the analytico-geometric perspective known as the method of characteristics (see, e.g. Boyd, 2001, Chap. 14) where the dynamics can be solved as a local system of ordinary differential equations, given by

$$\frac{\mathrm{d}x}{\mathrm{d}t} = u, \tag{31a}$$

$$\frac{\mathrm{d}c}{\mathrm{d}t} = 0. \tag{31b}$$

Each system of Eq. (31) describes the evolution of the couple $(x(t), c(t))$ starting from an initial position $x(0)$ where the scalar value is $c(0, x(0))$. At the geometric level, Eq. (31) remains to compute the trajectory of a mobile point of coordinate $x(t)$, the characteristic curve, solution of the dynamics (Eq. 31a) and transporting the scalar $c$ whose value $c(t)$ coincides with the field value $c(t, x(t))$. The transported value $c(t)$ evolves following Eq. (31b). In the present situation, since the right-hand side of Eq. (31b) is null, $c$ is conserved along the curve. This second point of view is referred to as the Lagrangian description for the transport.

Two discretization methods are interesting to study for the transport equation: the finite-difference approach and the semi-Lagrangian method resulting from the Lagrangian interpretation of Eq. (30).

The aim of this section is to detail the model-error covariance matrix for both schemes. This theoretical part is organized as follows. The error covariance parametric dynamics for the nature is first described considering the covariance model based on the diffusion equation; then both finite-difference and semi-Lagrangian schemes are introduced with their particular parametric dynamics.

### 3.1 PKF dynamics for the linear advection equation

To describe the time evolution of the predictability-error covariance matrix, Eq. (16), it is necessary to detail what is the TL dynamics for the linear transport, Eq. (30). Since this transport dynamics is linear, the error evolves according to the same dynamics, and the TL dynamics can be written as

$$\partial_t \widetilde{\varepsilon}^{\mathrm{p}} + u \partial_x \widetilde{\varepsilon}^{\mathrm{p}} = 0. \tag{32}$$

The PKF approximation of the forecast-error covariance matrix relies on the dynamics of the variance and of the diffusion fields deduced from Eq. (26). The equation for the variance is computed from Eq. (26a) by replacing the trend by the TL dynamics (Eq. 32), so that

$$\partial_t \widetilde{V}^{\mathrm{p}} = 2\mathbb{E}\left[\widetilde{\varepsilon}^{\mathrm{p}}\left(-u \partial_x \widetilde{\varepsilon}^{\mathrm{p}}\right)\right] = -2u\mathbb{E}\left[\widetilde{\varepsilon}^{\mathrm{p}} \partial_x \widetilde{\varepsilon}^{\mathrm{p}}\right]. \tag{33}$$

From $\partial_x (\widetilde{\varepsilon}^{\mathrm{p}})^2 = 2\widetilde{\varepsilon}^{\mathrm{p}} \partial_x \widetilde{\varepsilon}^{\mathrm{p}}$ and by the commutativity between the expectation operator and the spatial derivative, the variance dynamics becomes

$$\partial_t \widetilde{V}^{\mathrm{p}} = 2\mathbb{E}\left[\widetilde{\varepsilon}^{\mathrm{p}}\left(-u \partial_x \widetilde{\varepsilon}^{\mathrm{p}}\right)\right] = -u \partial_x \mathbb{E}\left[\left(\widetilde{\varepsilon}^{\mathrm{p}}\right)^2\right]. \tag{34}$$

By using the definition of the variance (Eq. 22), it results that the dynamics for the variance can be stated as

$$\partial_t \widetilde{V}^{\mathrm{p}} = -u \partial_x \widetilde{V}^{\mathrm{p}}. \tag{35}$$

The computation of the metric dynamics (Eq. 26b) is similar to the above computation made for the variance dynamics and is detailed in P16 and P18, where the interested reader is referred. It results that the PKF evolution for the nature is written as

$$\partial_t \widetilde{V}^{\mathrm{p}} + u \partial_x \widetilde{V}^{\mathrm{p}} = 0, \tag{36a}$$

$$\partial_t \widetilde{v}^{\mathrm{p}} + u \partial_x \widetilde{v}^{\mathrm{p}} = (2\partial_x u)\widetilde{v}^{\mathrm{p}}. \tag{36b}$$

Note that a similar system has been first obtained, in data assimilation, by Cohn (1993) (see their Eqs. 4.30a and 4.34 when written without stochastic model error).

From Eq. (36), it results that the variance and the diffusion are independent quantities. The variance is conserved while it is transported by the wind. The diffusion is not only transported, but it is also modified by the source term $(2\partial_x u)\widetilde{v}^{\mathrm{p}}$ which results from the deformation of correlations by the gradient of the flow $u$: the diffusion tensor is not conserved by the flow.

Hence, in this subsection, the predicability-error covariance for the nature (Eq. 16) has been computed for the linear transport (Eq. 30) and corresponds to the time integration of the uncoupled system of Eq. (36) starting from prescribed analysis-error variance and diffusion tensor fields.

The finite-difference scheme is now considered as a first numerical integration method for Eq. (30), with the derivation of the predictability-error covariance matrix.

### 3.2 Finite-difference scheme and its equivalent PKF dynamics

When the velocity field $u$ is positive (which is assumed from now without loss of generality), a conditionally stable discretization scheme is given by the Euler-upwind scheme:

$$\frac{c_i^{q+1} - c_i^q}{\delta t} = -u_i \frac{c_i^q - c_{i-1}^q}{\delta x}. \tag{37}$$

Stability is assured as long as the Courant–Friedrichs–Lewy (CFL) condition $\delta t < \delta x / \underset{x}{\mathrm{Max}} |u|$ is satisfied. Moreover, the scheme is consistent since in the limit of small $\delta x$ and $\delta t$, the dynamics (Eq. 30) is recovered from the discrete equation (Eq. 37). Thanks to the consistency and the stability, the equivalence theorem of Lax and Richtmyer (1956) assures to the convergence of Eq. (37) toward the true solution. Equation (37) stands as an illustration of model dynamics (Eq. 3).

While the numerical solution computed with the aid of a given numerical scheme can converge toward the true solution as $\delta t \to 0$ and $\delta x \to 0$, when $\delta t$ and $\delta x$ are of finite amplitude, the numerical solution often differs from the theoretical one. Actually, there exists another partial differential equation which offers a better fit to the numerical solution and highlights the properties of the numerical scheme (Hirt, 1968): the consistency, the stability as well as the dissipative and dispersive nature of the numerical scheme can be deduced from the so-called "modified equation" (Warming and Hyett, 1974). Hence, while it is supposed to solve Eq. (30), the numerical solution computed from Eq. (37) is actually the solution of the modified equation.

More precisely, if $C$ denotes a smooth function solution of the iterations (Eq. 37) with $C(q\delta t, i\delta x) = C_i^q$, then the modified equation is the partial differential equation verified by $C$ and at a given order of precision in $\delta t$ and $\delta x$. Here, it is straightforward to show that at order $\mathcal{O}(\delta t^2, \delta x^2)$, the partial differential equation best fitted by $C$ is given by (see Appendix C)

$$\partial_t C + U \partial_x C = \kappa \partial_x^2 C, \tag{38a}$$

where

$$U = u - \frac{\delta t}{2} \partial_t u + \frac{\delta t}{2} u \partial_x u \tag{38b}$$

and

$$\kappa = \frac{u}{2} (\delta x - u \delta t) \tag{38c}$$

are two functions of $t$ and $x$.

Compared with the nature (Eq. 30), the modified equation that best fits the Euler-upwind numerical scheme (Eq. 37) presents a correction of the wind which depends on the trend $\partial_t u$ and the self advection $u \partial_x u$ of the wind $u$. The magnitude of the correction scales as $\delta t$ and is null at the limit $\delta t \to 0$. But this is not the only modification of the dynamics, as a more critical difference emerges from the numerical discretization: a diffusion term whose magnitude depends on the CFL number $u \delta t / \delta x$. In particular, the diffusion coefficient is negative when the CFL number is larger than 1. The diffusion breaks the conservation property of the initial dynamics (Eq. 30). This example shows the importance of the modified equation: this provides a way to understand and characterize the defects due to the numerical resolution. In one dimension, for the evolution equation, this can be diffusive processes (associated with derivatives of even order) or dispersive processes (associated with derivatives of odd order).

From the PKF point of view, the modified equation is crucial since it converts a discrete dynamics into a partial differential equation, which appeared from P16 and P18, which is much simpler to handle when considering error covariance dynamics. Thanks to the modified equation (Eq. 38), it is now possible to compute the TL evolution of the predictability error for the Euler-upwind scheme, which can be expressed as

$$\partial_t \varepsilon^{\mathrm{p}} + U \partial_x \varepsilon^{\mathrm{p}} = \kappa \partial_x^2 \varepsilon^{\mathrm{p}}. \tag{39}$$

Equations of the PKF forecast can be computed under a similar derivation as in Sect. 3.1. To simplify the computation workflow, a splitting method has been introduced in P16 and P18. Due to the diffusion process appearing in Eq. (39), the PKF formulation faces a closure issue for which a closure scheme has been successfully proposed in P18: the Gaussian closure. The interested reader is referred to P18 for the details. Note that an alternative to the Gaussian closure can be deduced from the data through machine learning (Pannekoucke and Fablet, 2020). Hence, the resulting dynamics for the parameter of the predictability-error covariance model is given by

$$\partial_t V^{\mathrm{p}} + U \partial_x V^{\mathrm{p}} = -\frac{V^{\mathrm{p}} \kappa}{\nu^{\mathrm{p}}} + \kappa \partial_x^2 V^{\mathrm{p}} - \frac{\kappa (\partial_x V^{\mathrm{p}})^2}{2 V^{\mathrm{p}}} \tag{40a}$$

$$\partial_t \nu^{\mathrm{p}} + U \partial_x \nu^{\mathrm{p}} = (2 \partial_x U) \nu^{\mathrm{p}} + \kappa \partial_x^2 \nu^{\mathrm{p}} + 2\kappa$$

$$- \frac{2(\partial_x \nu^{\mathrm{p}})^2}{\nu^{\mathrm{p}}} \kappa + \partial_x \kappa \partial_x \nu^{\mathrm{p}} - \frac{2 \partial_x^2 V^{\mathrm{p}}}{V^{\mathrm{p}}} \kappa \nu^{\mathrm{p}} +$$

$$\frac{\partial_x V^{\mathrm{p}}}{V} \kappa \partial_x \nu^{\mathrm{p}} - \frac{2 \partial_x V^{\mathrm{p}}}{V^{\mathrm{p}}} \nu^{\mathrm{p}} \partial_x \kappa + \frac{2(\partial_x V^{\mathrm{p}})^2}{V^{\mathrm{p}2}} \kappa \nu^{\mathrm{p}}. \tag{40b}$$

Compared with the PKF dynamics of the nature (Eq. 36), the PKF for the Euler-upwind scheme gives rise to additional terms which result from the numerical diffusion of magnitude $\kappa$. Moreover, this time, the PKF for the Euler-upwind scheme presents a coupling between the variance and the diffusion, the coupling being a consequence of the numerical diffusion only. Note that a coupling between the variance and the correlation scale also appeared in Eqs. (4.30a) and (4.34) of Cohn (1993) but without a link to the discretization scheme.

The model-error covariance matrix, Eq. (27), associated with the Euler-upwind scheme can be deduced from the predictability-error covariance matrix approximations: starting from the initial analysis-error variance and diffusion field, integration of the parametric-error covariance equations of the nature (Eq. 36) and of the numerical discretization (Eq. 40) provides the predictability-error variances $\widetilde{V}^{\mathrm{p}}$ and $V^{\mathrm{p}}$, and the diffusion $\widetilde{\nu}^{\mathrm{p}}$ and $\nu^{\mathrm{p}}$, which are used to compute the model-error covariance parameter (Eq. 29).

As another example, the model-error parameters for the semi-Lagrangian scheme are now discussed.

### 3.3 Semi-Lagrangian scheme and its equivalent PKF dynamics

The modified equation technique has been previously considered for semi-Lagrangian (SL) schemes. For instance, McCalpin (1988) has shown for the case of constant advection velocity that a linear interpolation leads to an effective Laplacian dissipation, while the quadratic and cubic interpolations lead to a biharmonic dissipation.

Because we want to focus on the method to address the issue of the model error, and since uncertainty prediction of diffusive dynamics has been detailed by P18, we limit the presentation to the linear interpolation in the semi-Lagrangian scheme, and we present the modified equation of Eq. (30) for the study of its model error.

The Lagrangian perspective (Eqs. 31 to 30) suggests to build curves along which $c$ is constant. While simple, the drawback of this analytico-geometric method is the possible occurrence of curve trajectory collapses which prevent us from describing the time evolution of $c$ throughout the geographical domain. It is possible to take advantage of the geometrical resolution while avoiding the collapse by considering the so-called semi-Lagrangian procedure.

In the Lagrangian way of thinking, starting from a given position $x_o$, the question is where the mobile point lies along the time axis, which evolves the computation grid forward in time. The semi-Lagrangian perspective reverses this question by asking from which position $x_o^*$ originates the mobile point arriving at $x_o$ at a given time. Hence, the semi-Lagrangian scheme leaves the computation grid unchanged over the time steps of the integration while letting the scalar field $c$ evolve. More precisely for the particular dynamics of Eq. (30), by assuming the scalar field at time $t$ known for each points of the computational grid, for grid point $x_i$, the scalar field evolves as

$$c(t + \delta t, x_i) = c(t, x_i^*), \tag{41}$$

where $x_i^*$ is the origin of the trajectory at time $t$ which arrives at $x_i$ at time $t + \delta t$. Since the point of origin $x_i^*$ is unlikely to be a point of the computational grid (except for very particular situations), the value $c(t, x_i^*)$ is computed as an interpolation of the known values of $c$ at time $t$.

In its present form, the semi-Lagrangian procedure is not suited to the PKF method since it does not give rise any partial differential equation which lies at the core of the parametric approximation for covariance dynamics. To proceed further and to obtain PDEs, additional assumptions are introduced to translate the semi-Lagrangian procedure (Eq. 41) into a discrete scheme from which the modified equation is deduced.

In the case where the discretization satisfies the CFL condition $\delta t < \delta x / \underset{x}{\text{Max}} |u(x)|$ and for linear interpolation, it is straightforward to write the semi-Lagrangian procedure (Eq. 41) into a discrete scheme (see Appendix D for the de-

tails) which is stated as follows:

$$\begin{cases} \frac{c_i^{q+1} - c_i^q}{\delta t} = -u_i \frac{c_i^q - c_{i-1}^q}{\delta x}, & \text{for } u_i > 0 \\ \frac{c_i^{q+1} - c_i^q}{\delta t} = -u_i \frac{c_{i+1}^q - c_i^q}{\delta x}, & \text{for } u_i < 0 \end{cases} \tag{42}$$

which gives rise to the Euler-upwind/downwind schemes. Then, following the same derivation as previously presented in Sect. 3.2, the modified equation resulting from the scheme (Eq. 42) is given as the PDE verified by a smooth solution $C$ of Eq. (42). From the derivation detailed in Appendix D, the modified equation is

$$\partial_t C + U \partial_x C = \kappa^{\text{SL}} \partial_x^2 C, \tag{43a}$$

where

$$U = u - \frac{\delta t}{2} \partial_t u + \frac{\delta t}{2} u \partial_x u \tag{43b}$$

and

$$\kappa^{\text{SL}} = \frac{|u|}{2} (\delta x - |u| \delta t) \tag{43c}$$

are both functions of $t$ and $x$.

Hence, since this corresponds mainly to the modified equation (Eq. 38) encountered for the Euler-upwind scheme (Eq. 37), the parametric predictability-error covariance is also given by Eq. (40), replacing $\kappa$ by its SL counterpart value $\kappa^{\text{SL}}$.

Note that the derivation leading to the Euler-upwind and Euler-downwind schemes is due to the choice of the linear interpolation. The bridge between the SL and the Euler-upwind/downwind procedures is not a novelty. The derivation has been carried out since it offers an insight into how to build a modified equation for the SL scheme and also for the self-consistency of the presentation. In the general situation, the modified equation for the SL scheme is hard to obtain, if at all possible, and it is not the idea to claim the procedure as universal. But it provides a new insight into the model-error covariance matrix for the SL scheme, which is one of the main goals of the present contribution.

The next section presents the numerical experiments carried out to assess the ability of the PKF to characterize the model-error covariance matrix.

## 4 Numerical validation

### 4.1 Setting and illustration

In this experimental test bed, the domain is assumed to be the one-dimensional segment $[0, D)$ with periodic boundary conditions, where $D = 1$. The domain is discretized into a regular grid of $n = 241$ points $x_i = i \delta x$ for $i \in [0, 240]$ and $\delta x = D/n \approx 4.1 \, 10^{-3}$.

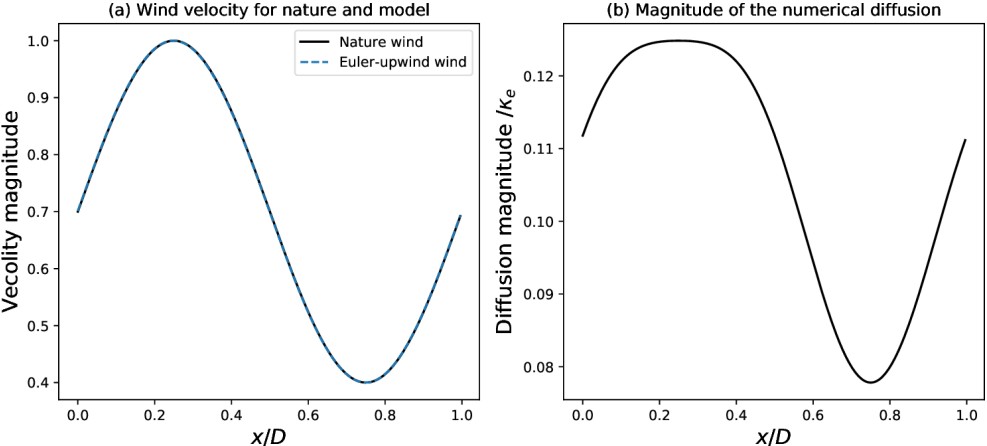

**Figure 2. (a)** Wind field specified for the nature dynamics and the one seen in the discretized model from Eq. (38b). Panel **(b)** represents the numerical diffusion coefficient due to the discretization equation (Eq. 38c), normalized by $\kappa_e = \delta x^2 / \delta t$.

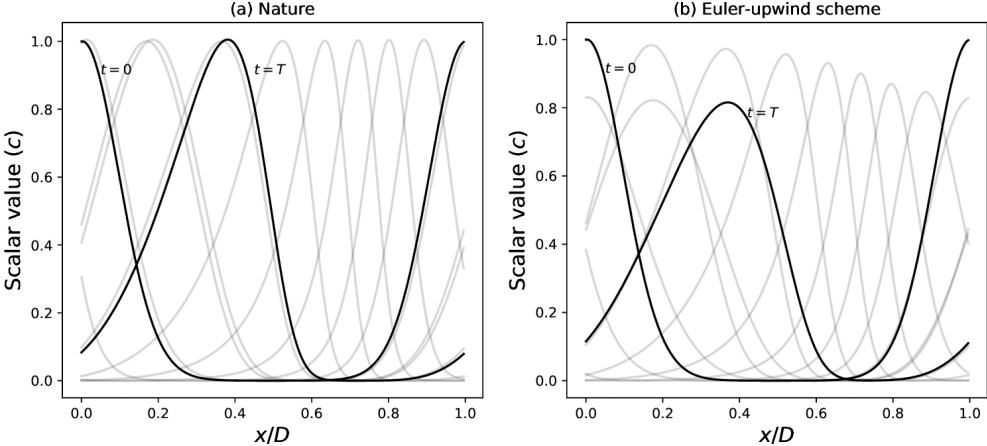

**Figure 3.** Nature **(a)** and numerical model **(b)** runs for times from $t = 0$ to $t = T$ and represented each $0.1T$.

The wind field $u$ for one-dimensional transport (Eq. 30) is set as the stationary field

$$u(x) = 0.4 + \frac{0.6}{2}\left(1 + \cos\left(\frac{2\pi}{D}(x - D/4)\right)\right), \tag{44}$$

shown in Fig. 2a, which appears as a jet with the entrance (exit) at $x = 0.75D$ ($x = 0.25D$): the flow accelerates (decelerates) until $x = 0.25D$ ($x = 0.75D$). For the latter, the lead time is $T = 2.0$.

In order to verify the CFL condition, the time step for the numerical simulation is set to $\delta t = 0.002$, leading to a CFL value of $0.48 < 1$. The magnitude of the numerical diffusion $\kappa$, Eq. (38c), associated with this setting is shown in Fig. 2b, normalized by the diffusion coefficient $\kappa_e = \delta x^2 / \delta t$.

For the numerical experiment, the initial state for $c$ is set to

$$c(0, x) = \exp\left(-\frac{1}{2(0.15D)^2}\sin\left(\frac{\pi}{2}(x - D/2)\right)^2\right), \tag{45}$$

while the initial analysis-error covariance matrix is set as the homogeneous Gaussian covariance matrix $\mathbf{P}_{t=0}^{\mathrm{f}}(x, y) = e^{-\frac{d(x,y)^2}{2l_h^2}}$ with $l_h = 0.05D \approx 12\delta x$, where $d(x, y) = \frac{D}{\pi}\left|\sin\frac{\pi}{D}(x - y)\right|$ is the chordal distance between the two geographical positions $x$ and $y$ (Pannekoucke et al., 2018a, see Eq. 3). The analysis-error standard deviation is set to the homogeneous value of 1.0.

For numerical validation, since no simple analytical solution of the partial differential equation (Eq. 30) exists, this dynamics is integrated considering a fourth-order Runge–Kutta time scheme applied on the finite-difference discretization:

$$\partial_t c_i = -u_i \frac{c_{i+1} - c_{i-1}}{2\delta x}, \tag{46}$$

where the spatial derivative is approximated by a centred second-order scheme. This constitutes the high-order discretization $\widehat{\mathcal{N}}$ of the nature $\mathcal{N}$, as introduced in Sect. 2.2: $\widehat{\mathcal{N}}$

is assumed to better reproduce the nature $\mathcal{N}$ than the model $\mathcal{M}$.

Figure 3 shows the trajectory computed from the nature approximated by $\widehat{\mathcal{N}}$ and the numerical model $\mathcal{M}$. Since the transport equation conserves the value of the field $c$, the extremal values of $c$ do not change along the integration and the wind $u > 0$ causes the initial structure to move to the right. While the field is conserved, it is also deformed by the wind. For the particular choice of the initial condition made here, the signal is of larger (smaller) scale in the region $x \in [0, 0.5]$ ( $x \in [0.5, 1]$) than its initial shape. Figure 3a shows that the nature approximation $\widehat{\mathcal{N}}$ is able to reproduce the conservation of $c$ as well as the stretching of the signal along the time axis. Hence, the nature approximation $\widehat{\mathcal{N}}$ is good enough to capture the main features of the nature dynamics, which justifies the use of this approximation in place of the true dynamics in the following. In contrast, the model $\mathcal{M}$ fails to maintain the magnitudes of the extrema (Fig. 3b), in accordance with the modified equation (Eq. 38a) of the Euler-upwind scheme (Eq. 37) which presents a non-physical diffusion process resulting from the numerical discretization. Note that the coefficient of the numerical diffusion is heterogeneous over the domain with a typical value of thereabout $0.1\kappa_e$ (see Fig. 2b). This heterogeneity is due to the scale variation of the signal, stretched by the wind shear: when the signal is of smaller (larger) scale to its initial shape, the second-order derivative is larger (smaller), which leads to an intensification (reduction) in the numerical diffusion term in Eq. (38a).

Having validated the two numerical models $\widehat{\mathcal{N}}$ and $\mathcal{M}$, it is now possible to look at the covariance dynamics and how the model-error covariance error can be estimated from the PKF prediction.

## 4.2 Assessment of the PKF in predicting the predictability-error covariance dynamics of the nature and of the numerical model

The PKF predictability-error covariance matrix dynamics for the transport equation (Eq. 30) is given by the system of Eq. (36). The PKF predictability-error covariance matrix dynamics resulting from the Euler-upwind integration (Eq. 37) is given by Eq. (40). Both systems are numerically integrated by considering, respectively, an explicit RK4 time scheme for the nature and an Euler time scheme for the Euler-upwind scheme. The time step used for the integration is $\delta t = 0.002$. The predictability-error variance fields are shown in Fig. 4. The predictability-error correlation length-scale fields, defined from the one-dimensional diffusion fields by $\widetilde{L}^{\mathrm{p}} = \sqrt{2\widetilde{\nu}^{\mathrm{p}}}$ (nature) and $L^{\mathrm{p}} = \sqrt{2\nu^{\mathrm{p}}}$ (numerical model), are shown in Fig. 5. The variance and the length scale are shown for the PKF and an ensemble estimation, the latter being only computed for the validation of the PKF (the ensembles are not needed nor used for the computation of the PKF systems).

To do so, an ensemble of $N_e = 6400$ analysis errors has been generated, $(\boldsymbol{\varepsilon}^{\mathrm{a}}_{0,k})_{k \in [1, N_e]}$, where each member is com-

puted as $\boldsymbol{\varepsilon}^{\mathrm{a}}_{0,k} = (\mathbf{P}^{\mathrm{f}}_{t=0})^{1/2} \zeta_k$ with $\zeta_k$ a sample of the Gaussian random vector of zero mean and covariance matrix of the identity matrix $\mathbf{I}$. This large size limits the sampling noise to a relative error of $1/\sqrt{N_e} \approx 1.25\%$.

Because the dynamics are linear, the TL nature and model are independent of any analysis state, and the ensemble is computed from the forecasts by the high-order discretization of the nature $\widehat{\mathcal{N}}$ and the model $\mathcal{M}$ of the ensemble of analysis errors $(\boldsymbol{\varepsilon}^{\mathrm{a}}_{0,k})$.

### 4.2.1 Validation of the PKF for the nature

The predictability-error covariance dynamics for the nature is first considered. Since the variance of the nature (Eq. 36a) is conserved, it results that, with the choice of an initial homogeneous variance, the trend is null and the variance field is the stationary homogeneous field (1.0). This theoretical result is well reproduced in Fig. 4a from the PKF integration, while the ensemble estimation (Fig. 4c) also shows this as stationary but within the sampling noise. The length scale (Fig. 5a) shows a periodic evolution where, starting from the homogeneous field of value $l_{\mathrm{h}}$, the length scale first increases (decreases) in the entrance (exit) of the jet; then these evolutions are attenuated and then compensated with the transport. Then ensemble estimation (Fig. 5c) presents the same variations (again to within the sampling noise), which validates the PKF dynamics for the nature. As a consequence, the PKF dynamics (Eq. 36) can be used to understand the dynamics of the uncertainty. In particular, the length-scale field at $t = 0.1T$ is well explained by the source/sink term $2(\partial_x u)\widetilde{\nu}^{\mathrm{p}}$ in Eq. (36b) whose magnitude, which lies between $-0.004$ and $0.004$, implies a rapid emergence of a heterogeneity leading to large (small) length scales for $x \in [0, 0.25D] \cup [0.75D, D]$ (for $x \in [0.25D, 0.75D]$), where $\partial_x u > 0$ ($\partial_x u < 0$); and by the transport term $u\partial_x \widetilde{\nu}^{\mathrm{p}}$ that shifts the fields to the right. Note that, by introducing the spatial average operator defined for any function $f$ by $\langle f \rangle(t) = \frac{1}{D} \int f(t, x) \mathrm{d}x$ as represented in Fig. 6, the averaged length scale $\langle \widetilde{L}^{\mathrm{p}} \rangle(t)$ ranges within $[12\delta x, 17.5\delta x]$ (see Fig. 6b), while the $\langle \widetilde{V}^{\mathrm{p}} \rangle(t)$ is a constant 1 (see Fig. 6b).

### 4.2.2 Validation of the PKF for the numerical model

The predictability-error covariance dynamics for the numerical model is now discussed. For the Euler-upwind scheme, the numerical diffusion resulting from the spatiotemporal discretization in Eq. (38a) implies a damping of the variance along the time axis (see Fig. 4b). The attenuation of the uncertainty governed by Eq. (40) leads to a heterogeneous damping over the domain and appears much stronger in the middle of the domain ($x = 0.5$) than near the boundaries ($x = 0$ and $x = 1$) while transported by the flow. The length scale (Fig. 5b) increases by the diffusion, while the shear produces similar patterns as for the forecast-error statistics. The ensemble estimation in Fig. 4d and Fig. 5d shows the same

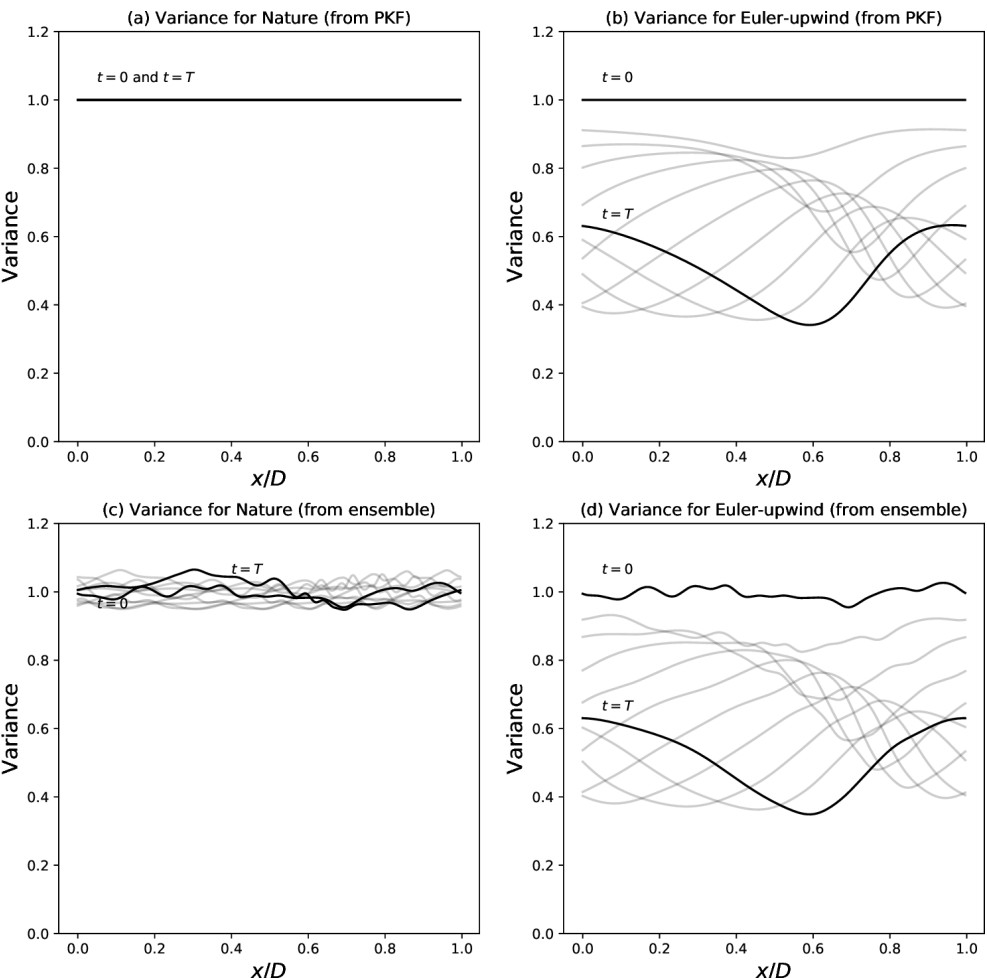

**Figure 4.** Predictability-error variance field, $\widetilde{V}^{\mathrm{p}}(t, x)$, for the nature (Eq. 30), computed from the PKF (Eq. 36) **(a)** and predictability-error variance field, $V^{\mathrm{p}}(t, x)$, for the numerical model resulting from the finite-difference and Euler discretization equation (Eq. 37), computed from the PKF (Eq. 40) **(b)**. Panels **(c)** and **(d)** are the ensemble estimation for panels **(a)** and **(b)**, where the nature dynamics is approximated by Eq. (46) dynamics in panel **(c)** (6400 members are used here). Fields are represented for times from $t = 0$ to $t = T$ and represented each $0.1T$.

signal as the PKF prediction (within the sampling noise) which validates the system of Eq. (40). As for the nature, it appears that the PKF dynamics for the numerical model, Eq. (40), explains the dynamics of the uncertainty. In particular, again, the length-scale field at $t = 0.1T$ is well explained by the source/sink strain term $2(\partial_x u)\widetilde{\nu}^{\mathrm{p}}$ in Eq. (40b) and by the transport term $u\partial_x \widetilde{\nu}^{\mathrm{p}}$, but this time, compared with Eq. (36b), the source term $2\kappa$ in Eq. (40b) implies an increase of the length scale $L^{\mathrm{p}}$. Note that the influence of the remaining terms in Eq. (40b) can be neglected at the prime instants of the dynamics: this is because at $t = 0$, $V^{\mathrm{p}}$ and $\nu^{\mathrm{p}}$ are constant fields ($V^{\mathrm{p}}(t = 0) = 1$ and $\nu^{\mathrm{p}}(t = 0) = l_{\mathrm{h}}^2/2$). Compared with the nature, the behaviour of the predictability-error variance of the numerical model presents some source/sink terms (right-hand side of Eq. 40a) that explain the emergence of a heterogeneity of the variance field. In particular, with the term $-\frac{\kappa}{\nu^{\mathrm{p}}} V^{\mathrm{p}}$ being strictly negative, it is responsible of the

damping of the variance; it is also responsible of the heterogeneity at the prime instants: with the length scale $L^{\mathrm{p}}$ being heterogeneous, the damping will be more (less) intense in the areas of small (large) length scales (compare Fig. 4b with Fig. 5b for $t = 0.1T$). In terms of spatial average, with the assumption that the variations around each averaged field are small so that for any fields $f$ and $g$ the approximation $\langle fg \rangle \approx \langle f \rangle \langle g \rangle$ applies, the spatial average of the dynamics (Eq. 40) is written as

$$\partial_t \langle V^{\mathrm{p}} \rangle = -\frac{\langle \kappa \rangle}{\langle \nu^{\mathrm{p}} \rangle} \langle V^{\mathrm{p}} \rangle, \tag{47a}$$

$$\partial_t \langle \nu^{\mathrm{p}} \rangle = 2\langle \kappa \rangle, \tag{47b}$$

where the property that for any function $f$ and integer $k > 0$, $\langle \partial_x^k f \rangle = 0$, has been used to eliminate all the other terms. Equation (47) can be solved analytically, and its solutions

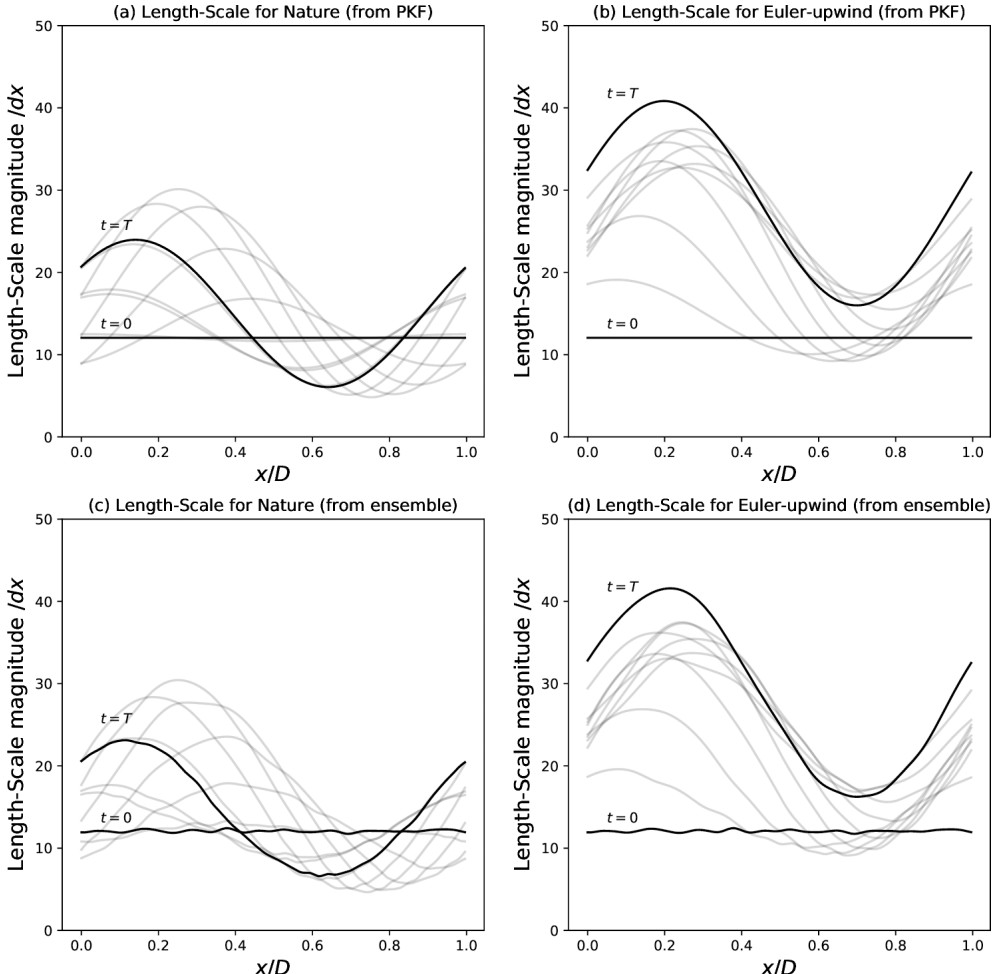

**Figure 5.** The length-scale counterpart of Fig. 4 representing the predictability-error length-scale fields $\widetilde{L}^p$ (nature) and $L^p$ (numerical model) in panels **(a, c)** (in panels **b–d**). The length scales are diagnosed from the diffusion coefficients from the formula $L = \sqrt{2\nu}$ and normalized by the grid spacing $\delta x$. Panels **(a)** and **(b)** are computed from the PKF, while **(c)** and **(d)** are estimated from the same large ensemble of forecasts as considered in Fig. 4. Fields are represented for times from $t = 0$ to $t = T$ and represented each $0.1T$.

are written as

$$\langle V^p \rangle(t) = \langle V^p \rangle(0) \left( \frac{\langle \nu^p \rangle(0)}{\langle \nu^p \rangle(0) + 2\langle \kappa \rangle t} \right)^{1/2}, \tag{48a}$$

$$\langle \nu^p \rangle(t) = \langle \nu^p \rangle(0) + 2\langle \kappa \rangle t. \tag{48b}$$

The analytical solution (Eq. 48) successfully reproduces the time evolution of the statistics in the present experiment. For the length scale, Eq. () reproduces the increase (see Fig. 6b), with an underestimation because this solution does not account for the oscillation due to the strain term that has been neglected in the dynamics (Eq. 47). For the variance, Eq. (48a) explains a linear decrease at the prime instant, followed by an attenuation in $t^{-1/2}$ (see Fig. 6a).

### 4.2.3 Intermediate result

As a conclusion of this section, the PKF appears able to predict the variance and the length-scale features of

the predicability-error covariance dynamics of the nature (Eq. 30) and of the numerical model, which corresponds to the discretization of the true dynamics given by Eq. (37). These results are now considered to provide an estimation of the model-error covariances.

### 4.3 Model-error covariance from the PKF prediction

From the previous section, the Euler-upwind discretization of the advection (Eq. 30) leads to a heterogeneous dissipative term, which affects the dynamics of the numerical model uncertainty by damping the variance while increasing the correlation length scale. When the bias due to the model error is lower than the predictability-error variance of the nature and the numerical model is dissipative, then the modelling (Eq. 21) of the model-error covariance matrix can be introduced, which is a flow-dependent modelling of the model-error covariance plus a climatological resid-

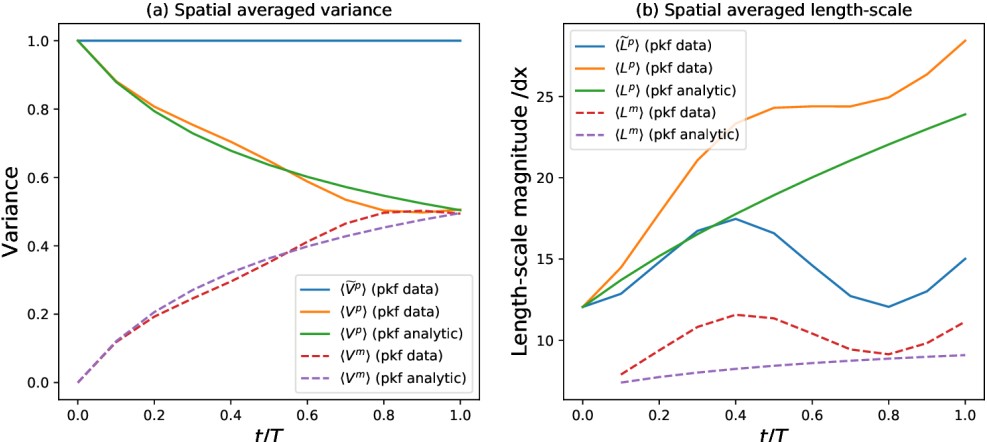

**Figure 6.** Time evolution of the spatial average over the domain of the predictability-error variance **(a)** and length scale **(b)**, computed from the PKF for the nature (blue) and the numerical model (orange). The analytical PKF approximation (Eq. 48) for the numerical model is in green. The model-error variance (Eq. 29a) and length scale (Eq. 49) are also represented (in dashed lines) for the spatial averaged of the PKF results shown in Fig. 7a and b (red), and the analytical approximation (purple).

ual. This is the situation encountered in the present numerical setting: the predictability-error variance of the nature is 1, which is larger than the bias (that is at most 0.2 when comparing the nature and the numerical model evolution in Fig. 3), while the predictability-error variance of the numerical model rapidly fails with, at its worst, a reduction of 60 % of the predictability-error variance of the nature (see the reduction at $x = 0.6D$ when comparing Fig. 4a and b). It results that the flow-dependent modelling (Eq. 21) may apply here.

In order to focus on the flow-dependent part of Eq. (21), the approximation (Eq. 27) is considered. Here, $\mathbf{P}^{\mathrm{m}}$ is computed from the parametric approach discussed in Sect. 2.4, with the parameters of Eq. (29), where the predictability-error covariance statistics are computed from Eq. (36) for the nature and Eq. (40) for the numerical model. Note that in this 1-D domain situation, Eq. (29b) is equivalent to the computation of the local correlation length scales by

$$L^{\mathrm{m}}(t, x) = \sqrt{\frac{V^{\mathrm{m}}}{\widetilde{V}^{\mathrm{p}}/(\widetilde{L}^{\mathrm{p}})^2 - V^{\mathrm{p}}/(L^{\mathrm{p}})^2}}. \tag{49}$$

The flow-dependent model-error covariance parameters are shown in Fig. 7, with the variance in panel (a) and the length scale in panel (b).

At the initial time, as there is no model error, the model-error variance is zero. But then, the model-error variance should increase linearly because the sink term $\frac{\kappa}{\nu^{\mathrm{p}}} V^{\mathrm{p}}$ that is the only non-zero right-hand-side term in Eqs. (36a) and (40a) (see also the spatially averaged dynamics; Eq. 47a) is a source of model-error variance at the initial time, so that for small $t$, the order of magnitude of $V^{\mathrm{m}}$ is given by

$$\langle V^{\mathrm{m}}\rangle(t) \sim t \frac{\langle \kappa \rangle}{\langle \nu^{\mathrm{p}} \rangle(0)} \langle V^{\mathrm{p}} \rangle(0), \tag{50}$$

which relates the increase of the model-error variance to the numerical diffusion. Note that the numerical diffusion is not the only process that induces a model error; e.g. the phase shift due to the correction of the numerical velocity $\frac{\delta t}{2} u \partial_x u$ in Eq. (38b) is also a source term, while it has been removed from by the averaging here. Hence, Eq. (50) provides the order of magnitude of the model-error variance at time $t = 0.1T$: when considering the initial conditions $\nu^{\mathrm{p}}(t = 0) = l_{\mathrm{h}}^2/2$ and $V^{\mathrm{p}}(t = 0) = 1$, and the order of magnitude of the diffusion coefficient $\langle \kappa \rangle \sim 0.1 \delta x^2/\delta t$ (see Fig. 2b), the typical value of the model-error variance computed from Eq. (50) is $\langle V^{\mathrm{m}}\rangle(0.1T) \sim 0.12$. This is in accordance with the typical values observed in Fig. 7a for that time. Note that the heterogeneity of the model-error variance field is due to the heterogeneity of the diffusion field $\nu^{\mathrm{p}}$ as discussed previously in Sect. 4.2.2.

Then, the model-error variance continues to grow, with a peak of uncertainty that evolves with the flow. In this numerical experiment, with the magnitude of the $\widetilde{V}^{\mathrm{p}}$ being constant and equal to 1, the magnitude of the model-error variance $V^{\mathrm{m}} = \widetilde{V}^{\mathrm{p}} - V^{\mathrm{p}}$, shown in Fig. 6a, evolves from Eq. (48a) as

$$\langle V^{\mathrm{m}}\rangle(t) \sim 1 - \left(\frac{l_{\mathrm{h}}^2}{l_{\mathrm{h}}^2 + 4\langle \kappa \rangle t}\right)^{1/2}, \tag{51}$$

when using the initial values $\langle \nu^{\mathrm{p}}\rangle(0) = \frac{1}{2}l_{\mathrm{h}}^2$ and $\langle V^{\mathrm{p}}\rangle(0) = 1.0$. Note that Eq. (51) asymptotically behaves as $1 - \frac{1}{2}\left(\frac{t}{\tau}\right)^{-1/2}$, where $\tau = \frac{l_{\mathrm{h}}^2}{\langle \kappa \rangle} \approx 1.3T$ is the half-magnitude time, which is in accordance with the simulation since $\langle V^{\mathrm{m}}\rangle(T) \sim 0.5$ at the end of the simulation.

The model-error length scale, given by Eq. (49), is more difficult to interpret (Fig. 7b) because of the oscillation due to the periodic domain. However, the evolution of the spatial average of the length-scale fields (dashed red line in Fig. 6b)

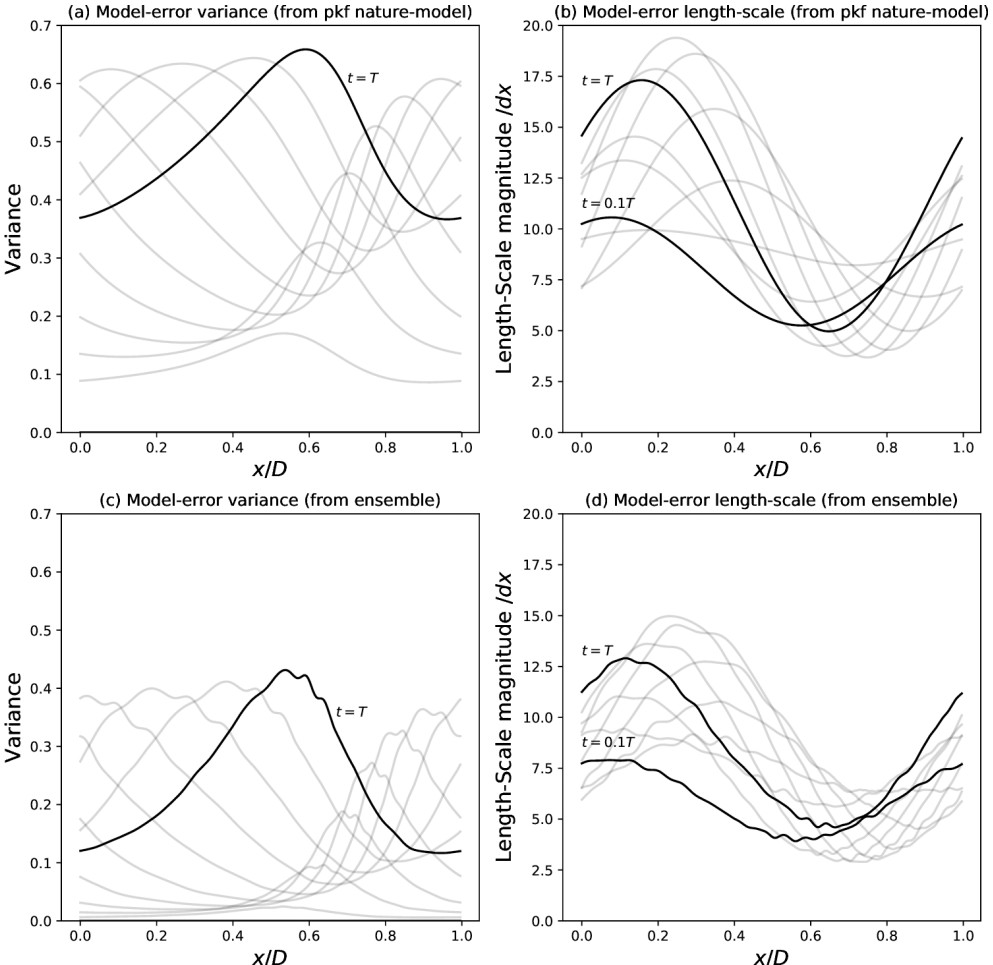

**Figure 7.** Flow-dependent model-error covariance, modelled from Eq. (21) as $\mathbf{P}^{\mathrm{m}} = \mathbf{\Pi}^{\mathrm{m}} + \mathbf{Q}$, and computed from the PKF for the nature and the Euler-upwind scheme. The variance **(a)** and the length scale (normalized by d$x$) **(b)** are represented for times from $t = 0.1T$ to $t = T$ at each $0.1T$ (for $t = 0$, the model error is null). Comparison with the ensemble estimation of the variance **(c)** and the length scale **(d)** of $\mathbf{P}^{\mathrm{ma}}$ (Eq. 53).

shows an increase of the averaged length scale with the time, which is in accordance with the order of magnitude for the model-error length scale (Eq. 49) computed from the analytical approximations (Eqs. 48 and 51), with $\langle \widetilde{V}^{\mathrm{p}} \rangle(t) = 1$ and $\langle \widetilde{L}^{\mathrm{p}} \rangle(t) \sim l_{\mathrm{h}}$ (dashed purple line in Fig. 6b).

Note that the model-error length scale is much smaller, but not null, which will balance the large length scale of the predictability-error covariance matrix $\mathbf{P}^{\mathrm{p}}$. Hence, as expected, the model error modelled by Eq. (21) is a heterogeneous covariance that depends on the state and the time: it is flow dependent.

It is interesting to compare $\mathbf{\Pi}^{\mathrm{m}}_{q+1}$ with the covariance of the unbiased error $\boldsymbol{\varepsilon}^{\mathrm{ma}}_{q+1} = (\mathbf{N} - \mathbf{M})\boldsymbol{\varepsilon}^{\mathrm{a}}_q$ that appears in the decomposition of the forecast error (see Eq. A3 in Appendix A)

$$\boldsymbol{\varepsilon}^{\mathrm{f}}_{q+1} = \boldsymbol{\varepsilon}^{\mathrm{p}}_{q+1} + \boldsymbol{\varepsilon}^{\mathrm{ma}}_{q+1} + \varepsilon^{\mathrm{m}}_{q+1}(\mathcal{X}^{\mathrm{a}}_q). \tag{52}$$

Indeed, if the errors on the right-hand side of Eq. (52) were decorrelated (which they are not), then $\mathbf{\Pi}^{\mathrm{m}}_{q+1}$ in Eq. (17)

would have been replaced by the covariance matrix $\mathbf{P}^{\mathrm{ma}} = \mathbb{E}\left[\boldsymbol{\varepsilon}^{\mathrm{ma}}_{q+1}(\boldsymbol{\varepsilon}^{\mathrm{ma}}_{q+1})^T\right]$ given by (see Eq. A5 in Appendix A)

$$\mathbf{P}^{\mathrm{ma}}_{q+1} = \mathbf{\Pi}^{\mathrm{m}}_{q+1} + \left[\left(\mathbf{MP}^{\mathrm{a}}_q \mathbf{D}^T\right) + \left(\mathbf{MP}^{\mathrm{a}}_q \mathbf{D}^T\right)^T\right], \tag{53}$$

with $\mathbf{D} = \mathbf{M} - \mathbf{N}$. In practice, $\mathbf{P}^{\mathrm{ma}}$ can be estimated from the ensemble of 6400 errors, $\boldsymbol{\varepsilon}^{\mathrm{ma}}_{q,k} = \widehat{\mathbf{N}}\boldsymbol{\varepsilon}^{\mathrm{a}}_{0,k} - \mathbf{M}\boldsymbol{\varepsilon}^{\mathrm{a}}_{0,k}$, where $\boldsymbol{\varepsilon}^{\mathrm{a}}_{0,k}$ is one of the analysis errors detailed in Sect. 4.2 and where $\widehat{\mathbf{N}}$ is the TL dynamics associated with the high-order numerical approximation $\widehat{\mathcal{N}}$ of $\mathcal{N}$. Because in the present experiment the dynamics of the nature and of the model are linear, $\boldsymbol{\varepsilon}^{\mathrm{ma}}_{q,k}$ is computed here as $\boldsymbol{\varepsilon}^{\mathrm{ma}}_{q,k} = \widehat{\mathcal{N}}(\boldsymbol{\varepsilon}^{\mathrm{a}}_{0,k}) - \mathcal{M}(\boldsymbol{\varepsilon}^{\mathrm{a}}_{0,k})$. The estimated variance and length-scale fields of $\mathbf{P}^{\mathrm{ma}}$ are shown in Fig. 7c and d. Compared with the PKF modelling (Fig. 7a and b), the time evolution shows a similar behaviour, but the variance of $\mathbf{P}^{\mathrm{ma}}$ is smaller, as well as its length scale. In this simulation, the contribution of the terms in $\mathbf{D}$, Eq. (53), is to reduce the

variance with a maximum of 0.4 at the end of the simulation. However, the minimum of variance of the predictability error is also nearly 0.4. Thus, if $\mathbf{P}^{\mathrm{ma}}$ was considered in place of $\mathbf{\Pi}^{\mathrm{m}}$, then a residual variance of order 0.2 would be needed (e.g. in $\mathbf{Q}$) so to obtain a magnitude of forecast error similar to the predictability of the nature.

Hence, the present numerical experiment illustrated and characterized the flow-dependent part of the model-error covariance $\mathbf{P}^{\mathrm{m}}$, modelled by Eq. (21), in the situation where the model error is related to the discretization of the advection by a heterogeneous wind, leading to a numerical model that is more diffusive than the nature. In this experiment, a linear increase in time, followed by a saturation in $t^{-1/2}$ has been found for the order of magnitude of the model-error variance. The residual climatological covariance, $\mathbf{Q}$ in Eq. (21), has yet to be estimated (not considered here).

## 5   Discussion

Before concluding, we end this work by addressing some general points about the flow-dependent model which has been introduced here.

The originality of the present contribution is two-fold. First, we have formulated a theoretical background corresponding to the model-error covariance matrix and introduced a modelling for its flow-dependent part, Eq. (21). This provides a theoretical framework to the correction of the predictability error introduced in M2000s. Then, we have provided theoretical and quantitative results about the diffusive effect due to the discretization that can lead to a loss of variance as observed in M2000s: this has been done by combining the formalism of the PKF and the modified equation. The interest for this modelling of the model-error covariance is supported by the results of M2000s, who have observed an improvement of the quality of the analysis in their data assimilation system of stratospheric observations.

The flow-dependent component of the model-error covariance introduced here can be computed in practice, because it relies on (1) the analysis uncertainty as characterized by the analysis state and its error covariance that can be estimated in data assimilation; and (2) the time evolution of the analysis-error covariance by the nature and by the numerical model that can be computed from an ensemble method or from the PKF approach.

Note that, if the difference between a low- and a high-resolution forecast is often used to compute the model-error at a given time, this does not tell anything about the model-error covariances at that time. At most, the model errors collected for a large number of dates, and for the same forecast time, can be used to compute the climatological bias and the climatological model-error covariance. To capture the error of the day following Eq. (21), the computation of the predictability-error covariance matrices is needed.

Hence, the use of the PKF is important because Eq. (21) needs to estimate not only the predictability-error covariance matrix of the numerical model but also the one of the nature. If an ensemble estimation of the latter matrix is possible in the research, e.g. by computing an ensemble of high-resolution forecast with $\widehat{\mathcal{N}}$ in place of the nature $\mathcal{N}$, it is too costly for real-time applications. It results that it is difficult to use Eq. (21) in an ensemble method. Compared with an ensemble method, the PKF remains to compute the evolution of a reduced set of covariance parameters by computing equations similar to the one encountered in geosciences. For the passive tracer in 1-D, the PKF dynamics consists in three equations: for the transport of the concentration, the dynamics of the variance and the dynamics of the local anisotropy (here a diffusion coefficient related to the correlation length scale). So, the numerical cost of the PKF (three equations) for the tracer (one equation) is about 3 times the computation of a single forecast compared to the dozens of members often used in ensemble methods (from which the statistics are corrupted by the sampling noise).

For the dynamics of a tracer, the PKF applies in 1-D as well as in 2-D and 3-D domains, where the number of equations are this time of five in 2-D and eight in 3-D (the additional equations are for the components of the local anisotropic tensor). However, in general, the use of the PKF is limited by the knowledge of the parameter dynamics. The formalism of the PKF is adapted for dynamics given by partial differential equations, as for the advection of a tracer, but the design of a multivariate PKF formulation is needed to address multivariate dynamics. Note that for the model error as presented here, the knowledge of the modified equation is a prerequisite that can be difficult to determine in general.

While the PKF is designed from the TL approximation, it is a second-order Gaussian filter that is a particular implementation of non-linear Kalman-like filters (Cohn, 1993): for non-linear dynamics, the PKF equation of the mean state depends on the second-order moments. However, for long-term predictions, or when the magnitude of the error is too large, the PKF would fail to provide an accurate estimation of the covariance matrices.

## 6   Conclusions

In this contribution, the part of the model-error covariance due to the spatiotemporal discretization scheme is explored by considering the parametric approximation for the Kalman filter (PKF). The PKF approach applies for a system whose dynamics is given by a set of PDEs. In the PKF formulation, covariances are approximated by covariance models characterized by a set of covariance parameters, whose dynamics is deduced from the PDEs of the system, supplemented by an appropriate closure if necessary. We focused on the class of covariance model distinguished by the variance field and the local anisotropic tensors (VLATcov). Therefore, for

VLATcov matrices, the covariance dynamics is given by the dynamics of the variance and the local anisotropic tensors, whose dynamics are deduced from the partial differential equations of the system.

In the case where the numerical model presents a dissipation due to the discretization, or where the numerical model is more dissipative than the nature, we introduced a modelling of the model-error covariance, where its flow-dependent part is approximated as the difference between the parametric approximation of the predictability-error covariance matrix of the nature and of the numerical model, plus a residual climatological covariance matrix. This modelling of the flow-dependent part can be computed in real applications because it relies on quantities that can be estimated: the analysis state and its analysis-error covariance matrix (or some of its characteristics). For a dynamics given by a partial differential equation, the parametric predictability-error covariance matrix of the nature is deduced from the evolution equation, while the predictability-error covariance matrix of the numerical model is computed from the modified evolution, i.e. the partial differential equations that best fits the numerical solution.

The ability of the parametric approach to characterize part of the model-error covariance dynamics has been illustrated in a numerical test bed in 1-D. We have considered the transport of a scalar by a heterogeneous velocity field. In this case, the parametric dynamics of the forecast error shows that the variance is conserved along the flow, while the local anisotropic tensor is transported by the flow and deformed by the gradient of the velocity.

For this transport dynamics, two numerical schemes have been considered: an Euler-upwind scheme and a semi-Lagrangian scheme in the case of a linear interpolation. The modified equations of both schemes make an additional heterogeneous dissipation and a perturbation of the velocity appear, whose characteristics depend on the spatiotemporal discretization ($dt$, $dx$), the trend and the shear of the flow. Because of the numerical diffusion, the variance of the predictability error is not conserved and a coupling with the anisotropy appears. This effect has been noted as well in 3-D global transport models (Ménard et al., 2020) where the loss of error variance is stronger for short correlation length scales.

An ensemble of forecasts has been introduced, taken as the reference, to compare the true covariance evolution with the parametric approximation. The numerical experiment shows the ability of the parametric dynamics to reproduce the predictability-error covariance dynamics. Then, the modelling of the flow-dependent part of the model-error covariance matrix has been computed and discussed. In particular, we discussed the growth of the model-error variance from the understanding of the PKF dynamics, showing a linear increase in time followed by a saturation in $t^{-1/2}$.

With the flow-dependent formulation being introduced for modelling the situation where the numerical model is more dissipative than the nature, the model-error variance provided by the PKF should be a lower bound of the true model-error variance, which needs a residual climatological covariance to account for the bias.

While there is no data assimilation experiment here, this contribution provides a theoretical background on the model-error covariance that sheds light on a study previously done by Ménard et al. (2000) and Ménard and Chang (2000) (M2000s), who have observed a loss of variance in the assimilation of a stratospheric tracer by using a Kalman filter: the variance forecasted was lower than the theoretical variance that is supposed to be conserved for the advection (Cohn, 1993). Actually, interpreted as an account of the model error due to the discretization scheme, the correction made by M2000s is similar to the modelling of the flow-dependent part of the model-error covariance matrix we proposed here. In particular, M2000s have observed that the Kalman filter, with the corrected predictability-error covariance, required less residual climatological model error (see Ménard et al., 2000, Sect. 5) and an improvement of the analysis-error statistics (see Ménard et al., 2000, Fig. 11), and thus indicated that the modelling of the model error, as proposed here, is in better agreement with optimality of the nature. Hence, the benefit of the flow-dependent modelling introduced here appears to be supported by the improvement of the analysis observed by M2000s in their experiment.

The methodology introduced here has shown the potential of exploring the model-error covariance from the parametric dynamics of error covariance. While the characterization of the model-error covariance is a challenge, as in air quality forecasts (Emili et al., 2016), the parametric approach appears as a new theoretical tool to tackle this issue. In order to represent the uncertainty of the small scales, it would be interesting to combine the parametric approach with other new methods, e.g. the modelling under location uncertainty (Resseguier et al., 2017).

However, the parametric dynamics faces closure issues that have to be addressed depending on applications. Here, the investigation of diffusive model errors has been made possible thanks to the Gaussian closure of P18. For other kind of numerical errors, an appropriate closure will have to be specified, either from theoretical closures or from the data as suggested by the data-driven and physics-informed identification of uncertainty dynamics of Pannekoucke and Fablet (2020).

## Appendix A: Expressions for the forecast error

The aim of this section is to provide the demonstrations of some decompositions of the forecast error: the usual expression as encountered in data assimilation, an expression where the model error is considered with respect to the analysis state and an expression that makes the predictability error appear with respect to the nature.

### A1   Expression of the forecast error as usually encountered in data assimilation

The forecast error is defined in Eq. (9) as the difference $\boldsymbol{\varepsilon}_{q+1}^{\mathrm{f}} = \mathcal{M}_{t_{q+1} \leftarrow t_q}(\mathcal{X}_q^{\mathrm{a}}) - \mathcal{X}_{q+1}^{\mathrm{t}}$. Thanks to Eq. (4), the true state at time $t_{q+1}$ can be replaced so that

$$\boldsymbol{\varepsilon}_{q+1}^{\mathrm{f}} = \mathcal{M}_{t_{q+1} \leftarrow t_q}(\mathcal{X}_q^{\mathrm{a}}) - \mathcal{M}_{t_{q+1} \leftarrow t_q}(\mathcal{X}_q^{\mathrm{t}}) + \varepsilon_{q+1}^{\mathrm{m}}(\mathcal{X}_q^{\mathrm{t}}), \quad \text{(A1)}$$

which makes the model error appear, defined by Eq. (5) as $\varepsilon_{q+1}^{\mathrm{m}} = \mathcal{M}_{t_{q+1} \leftarrow t_q} - \mathcal{N}_{t_{q+1} \leftarrow t_q}$. However, with $\mathcal{M}_{t_{q+1} \leftarrow t_q}(\mathcal{X}_q^{\mathrm{t}}) = \mathcal{M}_{t_{q+1} \leftarrow t_q}(\mathcal{X}_q^{\mathrm{a}} - \boldsymbol{\varepsilon}_q^{\mathrm{a}})$ which expands for small analysis error as

$$\mathcal{M}_{t_{q+1} \leftarrow t_q}(\mathcal{X}_q^{\mathrm{t}}) = \mathcal{M}_{t_{q+1} \leftarrow t_q}(\mathcal{X}_q^{\mathrm{a}}) - \mathbf{M}\boldsymbol{\varepsilon}_q^{\mathrm{a}},$$

(**M** denotes the propagator of the TL model along the analysis state trajectory; see Sect. 2.1 for details), the forecast error (Eq. A1) becomes

$$\boldsymbol{\varepsilon}_{q+1}^{\mathrm{f}} = \mathbf{M}\boldsymbol{\varepsilon}_q^{\mathrm{a}} + \varepsilon_{q+1}^{\mathrm{m}}(\mathcal{X}_q^{\mathrm{t}}),$$

which is written as

$$\boldsymbol{\varepsilon}_{q+1}^{\mathrm{f}} = \boldsymbol{\varepsilon}_{q+1}^{\mathrm{p}} + \varepsilon_{q+1}^{\mathrm{m}}(\mathcal{X}_q^{\mathrm{t}}), \quad \text{(A2)}$$

where $\boldsymbol{\varepsilon}_{q+1}^{\mathrm{p}} = \mathbf{M}\boldsymbol{\varepsilon}_q^{\mathrm{a}}$ is the predictability error (Eq. 12) with respect to the model. Equation (A2) is the expression of the forecast error usually introduced in data assimilation (Daley, 1992, see Eq. 2.8). Note that in this expression, the model error is evaluated at the true state $\mathcal{X}_q^{\mathrm{t}}$, while it is never known in practice. It would be interesting to consider an expression with known quantities, e.g. with the analysis state; this is now detailed in the next subsection.

### A2   Expression of the forecast error considering the model error with respect to the analysis state

The forecast error (Eq. A2) can be obtained by rewriting the model-error term as $\varepsilon_{q+1}^{\mathrm{m}}(\mathcal{X}_q^{\mathrm{t}}) = \varepsilon_{q+1}^{\mathrm{m}}(\mathcal{X}_q^{\mathrm{a}} - \boldsymbol{\varepsilon}_q^{\mathrm{a}})$. Hence, the Taylor expansion of $\varepsilon_{q+1}^{\mathrm{m}}$, with respect to $\mathcal{X}_q^{\mathrm{a}}$ for small error and lead time, leads to

$$\varepsilon_{q+1}^{\mathrm{m}}(\mathcal{X}_q^{\mathrm{t}}) = \varepsilon_{q+1}^{\mathrm{m}}(\mathcal{X}_q^{\mathrm{a}}) - d\varepsilon_{q+1,\mathcal{X}_q^{\mathrm{a}}}^{\mathrm{m}}\boldsymbol{\varepsilon}_q^{\mathrm{a}},$$

where $d\varepsilon^{\mathrm{m}}$ denotes the differential of the model error $\varepsilon^{\mathrm{m}} = \mathcal{M} - \mathcal{N}$ (Eq. 5) which exists when $\mathcal{M}$ and $\mathcal{N}$ are both differentiable, so that $d\varepsilon^{\mathrm{m}} = \mathrm{d}\mathcal{M} - \mathrm{d}\mathcal{N}$. It results that

$$\varepsilon_{q+1}^{\mathrm{m}}(\mathcal{X}_q^{\mathrm{t}}) = \varepsilon_{q+1}^{\mathrm{m}}(\mathcal{X}_q^{\mathrm{a}}) - (\mathbf{M} - \mathbf{N})\boldsymbol{\varepsilon}_q^{\mathrm{a}},$$

where **N** is the propagator of the TL nature along the analysis state trajectory (see Sect. 2.1 for details). Then, the forecast error (Eq. A2) expands as

$$\boldsymbol{\varepsilon}_{q+1}^{\mathrm{f}} = \boldsymbol{\varepsilon}_{q+1}^{\mathrm{p}} + \boldsymbol{\varepsilon}_{q+1}^{\mathrm{ma}} + \varepsilon_{q+1}^{\mathrm{m}}(\mathcal{X}_q^{\mathrm{a}}), \quad \text{(A3)}$$

where $\boldsymbol{\varepsilon}_{q+1}^{\mathrm{ma}}$ is defined by

$$\boldsymbol{\varepsilon}_{q+1}^{\mathrm{ma}} = (\mathbf{N} - \mathbf{M})\boldsymbol{\varepsilon}_q^{\mathrm{a}}. \quad \text{(A4)}$$

Note that $\boldsymbol{\varepsilon}_{q+1}^{\mathrm{ma}}$ is unbiased (at least when the analysis error is unbiased), i.e. $\mathbb{E}\left[\boldsymbol{\varepsilon}_{q+1}^{\mathrm{ma}}\right] = 0$, so that the covariance matrix is $\mathbf{P}_{q+1}^{\mathrm{ma}} = \mathbb{E}\left[\boldsymbol{\varepsilon}_{q+1}^{\mathrm{ma}}(\boldsymbol{\varepsilon}_{q+1}^{\mathrm{ma}})^T\right]$, which expands as

$$\mathbf{P}_{q+1}^{\mathrm{ma}} = \mathbf{N}\mathbf{P}_q^{\mathrm{a}}\mathbf{N}^T + \mathbf{M}\mathbf{P}_q^{\mathrm{a}}\mathbf{M}^T - \left[\left(\mathbf{N}\mathbf{P}_q^{\mathrm{a}}\mathbf{M}^T\right) + \left(\mathbf{N}\mathbf{P}_q^{\mathrm{a}}\mathbf{M}^T\right)^T\right].$$

Replacing the TL model **M** with $\mathbf{N} = \mathbf{M} - \mathbf{D}$ leads to

$$\mathbf{P}_{q+1}^{\mathrm{ma}} = \boldsymbol{\Pi}_{q+1}^{\mathrm{m}} + \left[\left(\mathbf{M}\mathbf{P}_q^{\mathrm{a}}\mathbf{D}^T\right) + \left(\mathbf{M}\mathbf{P}_q^{\mathrm{a}}\mathbf{D}^T\right)^T\right], \quad \text{(A5)}$$

where $\boldsymbol{\Pi}_{q+1}^{\mathrm{m}} = \mathbf{N}\mathbf{P}_q^{\mathrm{a}}\mathbf{N}^T - \mathbf{M}\mathbf{P}_q^{\mathrm{a}}\mathbf{M}^T$ (see Eq. 18).

As $\boldsymbol{\varepsilon}_{q+1}^{\mathrm{ma}}$ contains the predictability error, a final expression of the forecast error can be obtained as shown now.

### A3   Expression of the forecast error formulated in terms of nature predictability

Considering the definition of the predictability error (Eq. 12), the forecast error (Eq. A3) is rewritten as

$$\boldsymbol{\varepsilon}_{q+1}^{\mathrm{f}} = \mathbf{N}\boldsymbol{\varepsilon}_q^{\mathrm{a}} + \varepsilon_{q+1}^{\mathrm{m}}(\mathcal{X}_q^{\mathrm{a}}), \quad \text{(A6)}$$

which makes the predictability error appear with respect to the nature, $\widetilde{\boldsymbol{\varepsilon}}_{q+1}^{\mathrm{p}} = \mathbf{N}\boldsymbol{\varepsilon}_q^{\mathrm{a}}$.

Note that Eq. (A6) can be obtained directly from the definition of the forecast error (Eq. 9) as follows. By replacing the forecast with $\mathcal{M}_{t_{q+1} \leftarrow t_q}(\mathcal{X}_q^{\mathrm{a}}) = \mathcal{N}_{t_{q+1} \leftarrow t_q}(\mathcal{X}_q^{\mathrm{a}}) + \varepsilon_{q+1}^{\mathrm{m}}(\mathcal{X}_q^{\mathrm{a}})$, the forecast error is first written as $\boldsymbol{\varepsilon}_{q+1}^{\mathrm{f}} = \mathcal{N}_{t_{q+1} \leftarrow t_q}(\mathcal{X}_q^{\mathrm{a}}) - \mathcal{N}_{t_{q+1} \leftarrow t_q}(\mathcal{X}_q^{\mathrm{a}}) + \varepsilon_{q+1}^{\mathrm{m}}(\mathcal{X}_q^{\mathrm{a}})$, where the definition of the nature $\mathcal{X}_{q+1}^{\mathrm{t}} = \mathcal{N}_{t_{q+1} \leftarrow t_q}(\mathcal{X}_q^{\mathrm{t}})$ has been used. Then, rewriting $\mathcal{N}_{t_{q+1} \leftarrow t_q}(\mathcal{X}_q^{\mathrm{t}}) = \mathcal{N}_{t_{q+1} \leftarrow t_q}(\mathcal{X}_q^{\mathrm{a}} - \boldsymbol{\varepsilon}_q^{\mathrm{a}})$, whose Taylor expansion is $\mathcal{N}_{t_{q+1} \leftarrow t_q}(\mathcal{X}_q^{\mathrm{t}}) = \mathcal{N}_{t_{q+1} \leftarrow t_q}(\mathcal{X}_q^{\mathrm{a}}) - \mathbf{N}\boldsymbol{\varepsilon}_q^{\mathrm{a}}$, leads to the forecast error (Eq. A6).

## Appendix B: Approximation of the model-error metric tensor field

Here, we consider the particular case where the model-error covariance model is approximated as Eq. (27), i.e.

$$\mathbf{P}^{\mathrm{m}} \approx \boldsymbol{\Pi}^{\mathrm{m}} = \widetilde{\mathbf{P}}^{\mathrm{p}} - \mathbf{P}^{\mathrm{p}},$$

assuming this matrix is a covariance matrix. The local metric tensor can be diagnosed from the Taylor expansion of the model-error correlation function:

$$\rho^m(\mathbf{x}, \mathbf{x}+\delta\mathbf{x}) = \frac{1}{\sqrt{\mathbf{P}^m(\mathbf{x}, \mathbf{x})\mathbf{P}^m(\mathbf{x}+\delta\mathbf{x}, \mathbf{x}+\delta\mathbf{x})}}$$
$$\left(\widetilde{\mathbf{P}}^p(\mathbf{x}, \mathbf{x}+\delta\mathbf{x}) - \mathbf{P}^p(\mathbf{x}, \mathbf{x}+\delta\mathbf{x})\right). \tag{B1}$$

Under an assumption of local homogeneity of the variance, $\mathbf{P}^m(\mathbf{x},\mathbf{x}) \approx \mathbf{P}^m(\mathbf{x}+\delta\mathbf{x}, \mathbf{x}+\delta\mathbf{x})$, $\widetilde{\mathbf{P}}^p(\mathbf{x},\mathbf{x}) \approx \widetilde{\mathbf{P}}^p(\mathbf{x}+\delta\mathbf{x}, \mathbf{x}+\delta\mathbf{x})$ and $\mathbf{P}^p(\mathbf{x},\mathbf{x}) \approx \mathbf{P}^p(\mathbf{x}+\delta\mathbf{x}, \mathbf{x}+\delta\mathbf{x})$, which leads to the expansion

$$\rho^m(\mathbf{x}, \mathbf{x}+\delta\mathbf{x}) \approx \frac{\widetilde{\mathbf{P}}^p(\mathbf{x}, \mathbf{x})}{\mathbf{P}^m(\mathbf{x}, \mathbf{x})}\left(1 - \frac{1}{2}||\delta\mathbf{x}||^2_{\widetilde{\mathbf{g}}^p_{\mathbf{x}}}\right) -$$
$$\frac{\mathbf{P}^p(\mathbf{x}, \mathbf{x})}{\mathbf{P}^m(\mathbf{x}, \mathbf{x})}\left(1 - \frac{1}{2}||\delta\mathbf{x}||^2_{\mathbf{g}^p_{\mathbf{x}}}\right). \tag{B2}$$

Since $||\delta\mathbf{x}||^2_{\mathbf{g}_{\mathbf{x}}} = \delta\mathbf{x}^T \mathbf{g}_{\mathbf{x}}\delta\mathbf{x}$, the correlation is expanded as

$$\rho^m(\mathbf{x}, \mathbf{x}+\delta\mathbf{x}) \approx 1-$$
$$\frac{1}{2}\delta\mathbf{x}^T\left[\frac{1}{\mathbf{P}^m(\mathbf{x}, \mathbf{x})}\left(\widetilde{\mathbf{P}}^p(\mathbf{x}, \mathbf{x})\widetilde{\mathbf{g}}^p_{\mathbf{x}} - \mathbf{P}^p(\mathbf{x}, \mathbf{x})\mathbf{g}^p_{\mathbf{x}}\right)\right]\delta\mathbf{x}. \tag{B3}$$

After identification with the expected form of the expansion

$$\rho^m(\mathbf{x}, \mathbf{x}+\delta\mathbf{x}) \approx 1 - \frac{1}{2}||\delta\mathbf{x}||^2_{\mathbf{g}^m_{\mathbf{x}}}, \tag{B4}$$

it follows that

$$\mathbf{g}^m_{\mathbf{x}} = \frac{1}{\widetilde{V}^p(\mathbf{x}) - V^p(\mathbf{x})}\left(\widetilde{V}^p(\mathbf{x})\widetilde{\mathbf{g}}^p_{\mathbf{x}} - V^p(\mathbf{x})\mathbf{g}^p_{\mathbf{x}}\right), \tag{B5}$$

where the variance is denoted by $\widetilde{\mathbf{P}}^p(\mathbf{x}, \mathbf{x}) = \widetilde{V}^p(\mathbf{x})$ and $\mathbf{P}^p(\mathbf{x}, \mathbf{x}) = V^p(\mathbf{x})$.

## Appendix C: Computation of the modified equation for the Euler scheme

The modified partial differential equation associated with the numerical scheme (Eq. 37) is the partial differential equation of a smooth function $C$, solution of the scheme, so that $C(q\delta t, i\delta x) = C^q_i$, i.e.

$$\frac{C^{q+1}_i - C^q_i}{\delta t} = -u_i\frac{C^q_i - C^q_{i-1}}{\delta x}, \tag{C1}$$

for which the Taylor formula in time and space on the order of $\mathcal{O}(\delta t^2, \delta x^2)$ is

$$\partial_t C + \frac{\delta t}{2}\partial^2_t C + \mathcal{O}(\delta t^2) =$$
$$-u\left(\partial_x C - \frac{\delta x}{2}\partial^2_x C + \mathcal{O}(\delta x^2)\right). \tag{C2}$$

The second-order time derivative can be replaced from the equation (Eq. C2) itself, at an appropriate order. Due to the

$\delta t$, an expansion at order $\mathcal{O}(\delta t)$ only requires to express the second-order derivative at the lead order, which is from

$$\partial_t C = -u\partial_x C + \mathcal{O}(\delta t, \delta x). \tag{C3}$$

Then, from the time derivative, the second-order derivative can be replaced by

$$\partial^2_t C = \partial_t\left(-u\partial_x C\right) + \mathcal{O}(\delta t, \delta x),$$
$$= -\partial_t u\partial_x C - u\partial^2_{xt} C + \mathcal{O}(\delta t, \delta x).$$

Consequently, the second-order derivative $\partial^2_{xt} C$ can be deduced from spatial derivative of Eq. (C3) and is written as

$$\partial^2_{xt} C = -\partial_x(u\partial_x C) + \mathcal{O}(\delta t, \delta x),$$
$$= -\partial_x u\partial_x C - u\partial^2_x C + \mathcal{O}(\delta t, \delta x).$$

It results that Eq. (C2) is written as

$$\partial_t C + \frac{\delta t}{2}\left[-\partial_t u\partial_x C - u\left(-\partial_x u\partial_x C - u\partial^2_x C\right)\right] =$$
$$-u\left(\partial_x C - \frac{\delta x}{2}\partial^2_x C\right) + \mathcal{O}(\delta t^2, \delta x^2),$$

so that

$$\partial_t C + U\partial_x C = \kappa\partial^2_x C + \mathcal{O}(\delta t^2, \delta x^2), \tag{C4}$$

where $U = u - \frac{\delta t}{2}\partial_t u + \frac{\delta t}{2}u\partial_x u$ and $\kappa = \frac{u}{2}(\delta x - u\delta t)$ are two functions of $t$ and $x$.

## Appendix D: Computation of the modified equation for a semi-Lagrangian scheme

The aims of this section are two-fold: the first goal is to obtain a discrete scheme from the semi-Lagrangian procedure, and the second goal is to deduce the modified equation of the discrete scheme.

For the sake of simplicity, the linear advection dynamics $\partial_t c + u\partial_x c = 0$ is first considered with a velocity $u > 0$.

From the characteristic curve resolution, it follows that $c(t_{q+1}, x_i) = c(t_q, x^*_i)$, where the originating point $x^*_i$ is assumed in between points $x_{i-1}$ and $x_i$, which means that the CFL constraint $u\delta t < \delta x$ is verified. This originating point can be approximated as $x^*_i = x_i - u_i\delta t$, and if a linear interpolation is considered for the computation of $c(t, x^*_i)$, it follows that

$$c(t_q, x^*_i) = \left(1 - \frac{x^*_i - x_{i-1}}{x_i - x_{i-1}}\right)c^q_{i-1} + \left(\frac{x^*_i - x_{i-1}}{x_i - x_{i-1}}\right)c^q_i \tag{D1}$$
$$= \frac{u_i\delta t}{\delta x}c^q_{i-1} + \left(1 - \frac{u_i\delta t}{\delta x}\right)c^q_i. \tag{D2}$$

Hence, the numerical scheme is written as

$$c_{q+1, i} = \frac{u_i\delta t}{\delta x}c_{q, i-1} + \left(1 - \frac{u_i\delta t}{\delta x}\right)c_{q, i}. \tag{D3}$$

The modified differential equation is obtained by replacing $c$ by a smooth function $\tilde{c}$, the solution of the numerical scheme (Eq. D3). The computation of the modified equation is similar to the Euler case detailed in Appendix C, leading to

$$\partial_t C + \left( u - \frac{\delta t}{2} \partial_t u + \frac{\delta t}{2} u \partial_x u \right) \partial_x C = \left( \frac{1}{2} u \delta x - \frac{1}{2} u^2 \delta t \right) \partial_x^2 C. \tag{D4}$$

When $u < 0$, the differential equation is written as

$$\partial_t C + \left( u - \frac{\delta t}{2} \partial_t u + \frac{\delta t}{2} u \partial_x u \right) \partial_x C = \left( \frac{1}{2} (-u) \delta x - \frac{1}{2} u^2 \delta t \right) \partial_x^2 C. \tag{D5}$$

Hence, in the general situation,

$$\partial_t C + \left( u - \frac{\delta t}{2} \partial_t u + \frac{\delta t}{2} u \partial_x u \right) \partial_x C = \left( \frac{1}{2} |u| \delta x - \frac{1}{2} u^2 \delta t \right) \partial_x^2 C, \tag{D6}$$

whatever the sign of the velocity $u$.

*Data availability.* The data have been generated from a numerical experiment as described in the paper. TS4

*Author contributions.* OP and RM conceived the idea to explore the influence of the numerical scheme on the error model. OP linked the modified equation to the parametric formulation of the uncertainty prediction. MEA contributed to the simulation during its training period, supervised by OP and MP.

*Competing interests.* The authors declare that they have no conflict of interest.

*Acknowledgements.* We would like to thank Mateusz Reszka for English proofreading. We would like to thank the three anonymous referees for their fruitful comments which have contributed to improving the manuscript.

*Financial support.* This research has been supported by the LEFE INSU (KAPA grant).

*Review statement.* This paper was edited by Wansuo Duan and reviewed by three anonymous referees.

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

TS7 Please confirm.