# Peer review of "A methodology to obtain model-error covariances due to the discretization scheme from the parametric Kalman filter perspective"

_Nonlinear Processes in Geophysics, 2020_

## Referee Comment (RC1) · Anonymous Referee #1 · 28 May 2020

**1   General comments**

This paper is a new and very interesting piece of work for the geophysical data assimilation community.

The authors propose a way to include spatial-discretization (model) error in the parametric Kalman filter (PKF) framework. In this aim, the model error covariance is approximated by the difference between the predictability error covariance and the forecast error covariance. Predictability and forecast errors are associated to the erroneous and perfect models respectively, the erroneous model being a spatial discretization of the

perfect model. Using an almost equivalent partial derivative equation for the erroneous model and the Gaussian closure of Pannekoucke et al. (2018), the authors derive evolution laws for predictability and forecast errors, their PKF representations (variance and diffusion tensor) and finally the model error PKF representation.

Representing spatial-discretization error, the use of modified equations (32) for this purpose and the combination with PKF are very good ideas and seems to be promising paths. Nevertheless, I was confused by the predictability and forecast errors definitions and by the ensuing model error proxy.

**2  Specific comments**

- Some parts of section 2.1 are unclear to me, especially between lines 85 and 90. The forecast $\chi^f$ (and thus the forecast error $\epsilon^f$) is first defined by the perfect model $\mathcal{N}$ (eq. (4)). Then, it is re-defined by the erroneous model $\mathcal{M}$ (eq. (11)). I understand that you wanted to introduce the error complexity step by step. But, I think that lines 85 to 90 confuse the reader since in the following of the paper we are not sure of what is your definitions of the forecast $\chi^f$ and forecast error $\epsilon^f$.

- I am particularly confused by the equation of the line 88 :

$$\epsilon^f = M\epsilon^a - \epsilon^m, \tag{1}$$

which leads to the central proxy equation (13) (and then leading to (15), (16), (17), ...). If the analysis error is zero, then the forecast error $\epsilon^f$ should be equal to $0$ according to the definition (2)-(5) and this would imply zero model error $\epsilon^m$. However, model error can exist even with perfectly known initial condition. In my understanding,

$$\epsilon^f = N\epsilon^a, \tag{2}$$

$$= M\epsilon^a + (N - M)\epsilon^a, \tag{3}$$

$$= M\epsilon^a + \int_0^t (\mathcal{N} - \mathcal{M})(\chi^t + \epsilon^a) - \int_0^t (\mathcal{N} - \mathcal{M})(\chi^t), \tag{4}$$

$$\neq M\epsilon^a - \int_0^t (\mathcal{N} - \mathcal{M})(\chi^t), \tag{5}$$

$$= M\epsilon^a - \epsilon^m. \tag{6}$$

The model error source term, $\int_0^t (\mathcal{N} - \mathcal{M})(\chi^t + \epsilon^a) = \int_0^t (\mathcal{N} - \mathcal{M})(\chi^a)$, is missing. I would expect to see this source term, at least later, in the draft, like in the equation (33). For the example 3.2, this source term corresponds to the residual of the spatial discretization:

$$(\mathcal{N} - \mathcal{M})(\chi^a) = [(v - U) \cdot \nabla - \kappa\Delta] \chi^a \tag{7}$$

Intuitively, I thought that this term should appear somewhere to increase the model error along time (especially if ones neglects the correlation between analysis error and model error). But, I may miss something. Perhaps, the authors do not consider it because they consider it as a bias and not as a centred random error? or because this source term is negligible? Is it possible to quantify its relative order of magnitude? If the source term is negligible, how can the independent-from-analysis-error model error component can growth? If the source term is not negligible but not considered by the authors because it is a bias, perhaps the source term should be replaced by a centered noise having the same effect (eg the same norm)? The authors should discuss these points.

- From a more general perspective, if ones focuses on the part of the model error which is independent of the analysis error, I guess that the methodology should work with zero analysis error. Am I right? Can your method deal with zero analysis error?

- I am not sure to understand what is the predictability error $\epsilon^p$. I guess it is just defined equation (14). So, in the linear approximation, $\epsilon^p$ is the propagation in time of the analysis error $\epsilon^a$ with the erroneous model $\mathcal{M}$. Am I right? If not, it may alter my previous comments but then I would not understand the equation (33).

**3  Technical corrections and typos**

- Several equations appeared separated in two, like in lines 110 and 264. Perhaps, this is just due to the draft template.

- line 227 : trom $\mapsto$ from

- line 255 : a space is missing at the end of the sentence
* * *

---

## Referee Comment (RC2) · Anonymous Referee #2 · 22 Jun 2020

1, Please discuss the condition that the assumption of the decorrelation between the analysis error and the model error is valid. 2, Since the prediction error is obtained from the tangent linear model, its validity depends on the error magnitude and the forecast range. Thus the method proposed in this paper also depends on the error magnitude and the forecast range? Please discuss. 3, There are many formulas in the paper, please make sure all the symbols have been explained. For example, what is the meaning of subscript "q"? 4, To what extend can the PKF approximation provide the estimation of model-error covariance characteristics? Will it depend on the complexity of the model?

---

## Referee Comment (RC3) · Anonymous Referee #3 · 1 Jul 2020

**1   General response and main points**

Overall I feel positive about the manuscript. The analytical work provides some interesting interpretations on the estimation of model error covariances. The numerical demonstrations are also interesting. However, there are a number of portions of the manuscript that can use clarification and greater consistency before it is ready for publication. Please see the main points to address in revision below, minor points are covered in section 2.

1. Equations throughout the discussion version are poorly formatted and awkwardly

split, this should be fixed for better readability. This could also use hyperlinking in equation references to make the document more easily searchable.

2. I agree with the other review comments that the definitions of quantities such as the model error $\epsilon_{q+1}^m$, and their evolution equations, need to be more clear.

3. My understanding is that the CFL condition is stated as, $C = \frac{u\delta_t}{\delta_x} \leq C_{max}$ where $C_{max}$ is typically taken equal to $1$. The CFL condition is stated differently in lines 219, page 8, and 301, page 11, and it does not seem to me that these statements correspond to the above condition. Please check these lines for consistency.

4. In appendix B, lines 495 - 499, page 21: when going through the derivation of the modified equation for the Euler scheme, I find that $\kappa = \frac{u}{2}\left(\delta_x + u\delta_t\right)$ as the term $\frac{-u^2\delta_t}{2}\partial_x^2\tilde{c}$ appears on the left-hand-side of the equation on line 495. Please verify the equations for consistency and update the discussion in lines 238 - 246, page 9. Does this affect any of the numerical results such as Fig 1.b? Are there any other places where this would change the interpretation of the results?

5. The conclusion of the numerics is too short. The main point of the work, estimating the model error covariance via the PKF formalism is somewhat successful, and the limitation of the decorrelated error assumption is discussed. However, there isn't much quantitative analysis of practical use of this model error estimate. For instance, can the estimated variance in the PKF formalism be used as an upper bound for the true model error variance?

**2 Minor Points**

1. Page 5, lines 121 - 123: I find the parentheses in this sentence to be awkward, as it is easy to confuse their use as mathematical arguments. The conditional

statements for the forecast, respectively the analysis, should use commas to clarify the meaning of the text.

2. Page 10, line 255: see the typo "details.Note".

3. Page 13, lines 343 - 344: the definition of $\widehat{\mathcal{N}}$ needs to be clarified. Currently the sentence doesn't seem to be complete. I find the notations confusing here, where $\mathcal{N}$ is referred to as a model equation. In this context, what equations represent $\mathcal{M}$?

4. Page 15, line 385: see typo "legnth-scale".

5. Page 21, Eq. C1: see typo $x_{ki-1}$.

---

## Author Comment (AC1) · 14 Sep 2020

Final answer to the referee 1

First of all, we would like to thank the referee for her/his review on our paper and for giving us the opportunity to improve our paper.

Now, we organized the answer to the comments as follows. First, we list some changes afford to the manuscript then detail our answers to the questions raised by the referee.

**List of changes for the revision**

*Major changes*

The formalism has been reworded to match the usual notation of data assimilation (Sec. 2). In particular Sec. 2.1 has been split so to better introduce the modelling of the model-error covariance matrix that is now presented in Sec. 2.2. To this end, a new figure 1 has been introduced to sketch the dynamics of the uncertainty in presence of model error and to facilitate the setting of the framework considered here: the case where the numerical model is (more) dissipative compared to the nature. The formalism of the model error has been clarified, with the details of formulations of the forecast error that are given in Appendix A. A specific notation has been introduced for the predictability error of the nature ($\widetilde{\eps}^p$) so to avoid any confusion with the forecast error as usually encountered in data assimilation. Now, every $\widetilde{.}$ symbols refer to the uncertainty of the nature.

Sec 4.2 has been split in two part, one for the nature, the other for the model. The dynamics of the average over the model has been introduced to quantity the evolution of the variance and the length-scale fields. In particular, the Sec. 4.2.2 provides an analytical evolution of the predictability-error variance of the model which better explore the PKF equations. This quantitative evolution of the predictability-error variance is also now explored in section 4.3 for the model-error variance.

A section 5 has been introduced to discuss the results and the connexion with the previous work of Ménard et al 2000s that support the interest of the method introduced here, but in a real data assimilation system.

*Minor changes*

The detail of the ensemble estimation was missing in the previous version of the manuscript, this has been introduced in the introduction of Sec. 4.2.

There was an error on the definition of the time representation of Fig. 3 and later where the results are shown every $0.1T$, moreover the lead time of the simulation has been indicated $T=2.0$ (this was missing).

P10, line 12: The chordal distance was used in the numerical simulation, but we wrote the distance as |x-y| in the manuscript. While there is no difference between the two distances with the value of D and l_h used here, the chordal distance leads to a valid correlation function on the circle (that is not the case with the previous distance – this theoretical consideration is not discussed in the manuscript but justifications can be found in the article Pannekoucke et al. 2018).

*Differences between the two version of the manuscript*

To facilitate the comparison between the two version of the manuscript, a companion version of the manuscript lists all the modifications where old (new) statements are in red (blue).

**Answer to the question of the referee**

We copied your commentary in italics below, we reply in normal blue font.

*1. "Some parts of section 2.1 are unclear to me, especially between lines 85 and 90. The forecast χ f (and thus the forecast error $\eps^f$ ) is first defined by the perfect model N (eq. (4)). Then, it is re-defined by the erroneous model M (eq. (11)). I understand that you wanted to introduce the error complexity step by step. But, I think that lines 85 to 90 confuse the reader since in the following of the paper we are not sure of what is your definitions of the forecast χ f and forecast error # f"*

The section has been rephrased and the presentation of the model error has been clarified. In particular, in the previous version of the manuscript, we considered that the true forecast-error covariance matrix was the predictability-error of the nature which leads to a confusion in the presentation. Now, the formalism of the data assimilation, and especially the one of the model-error covariance is clearly stated, and a specific new Sec 2.2 has been introduced to present the way we understand the specification of the model-error, as a modelling of the flow-dependent part of the model-error covariance. This time, we only compared the predictability-error covariance of the nature and of the numerical model. Hence, the forecast-error is now uniquely defined by Eq.(9), where the forecast is the usual definition given in Eq. (8).

*2. "I am particularly confused by the equation of the line 88 [..] which leads to the central proxy equation (13) (and then leading to (15), (16), (17), ...). If the analysis error is zero, then the forecast error e^f should be equal to 0 according to the definition (2)-(5) and this would imply zero model error e^m. However, model error can exist even with perfectly known initial condition. [..] The model error source term [..] is missing. [..] Intuitively, I thought that this term should appear somewhere to increase the model error along time (especially if ones neglects the correlation between analysis error and model error). But, I may miss something. Perhaps, the authors do not consider it because they consider it as a bias and not as a centred random error? or because this source term is negligible? Is it possible to quantify its relative order of magnitude? If the source term is negligible, how can the independent-from-analysis-error model error component can growth? If the source term is not negligible but not considered by the authors because it is a bias, perhaps the source term should be replaced by a centered noise having the same effect (eg the same norm)? The authors should discuss these points."*

The revised version of the formulation and of the presentation of the assimilation should now answer to the referee's comment while avoiding any confusion. Moreover the Appendix A now detail how the forecast error is related to the model error with respect to the true state (Eq. 10) and to the analysis state (Eq. 19) which clarify the manuscript.

In the revised version of the manuscript, the modelling of the model-error covariance Eq.(16) includes the situation, suggested by the referee, where the initial state is perfectly known: in that case, the flow-dependent part Pi^m is zero, but the covariance matrix is not null as it equals to the climatological residual part Q. Note that there is no bias term in Eq. 39 (Eq. 33 in the previous manuscript) -- as suggested by the referee -- because this corresponds to the predictability error dynamics. However we agree that a source term independent on the analysis error exists, a reference to Nicolis 2003 Eq.(4) is indicated to tackle this situation, while it is not addressed in this contribution (the effect of the source term is in $Q$) (see Section 2.2, p 5, l13):

"Note also that, in panel (b), assuming that there is no bias when there is one leads to consider the

bias as a residual model-error that could be estimated from the climatology. Hence, \
Eq{eq_model_Pm} is an hybrid model that balance the model error of the day with the climatological effect of the model error. In particular, if the initial state is perfectly known, the $\mathbf{\Pi}^m_{q+1}$ is zero, and the model-error is then characterized by the climatological residual term $\mathrm{Q}_{q+1}$: the source of this uncertainty corresponds to a forcing term that appears in the dynamics of the forecast error (see e.g. Eq. (4) in Nicolis (2003)) ; this source term is not explored here and its contribution is incorporated in Q whose magnitude depends on the lead time."

In the new version of the manuscript, we clarified the presentation of the flow-dependent modelling for P^m by introducing a sketch of the dynamics of the uncertainty (Fig. 1) and then theoretical justifications that make appears a drift given by the model error with respect to the analysis state. In particular, we focused in the flow-dependent part of the P^m where the bias has been neglected (see p6,l 45):

"From now, we will assume that Π^m is a covariance matrix, and that there is no residual model-error Q so to focus on Π^m alone, so that P^m_{q+1} ≈ Π^m _{q+1} (Eq. 27)"

3. *"From a more general perspective, if ones focuses on the part of the model error which is independent of the analysis error, I guess that the methodology should work with zero analysis error. Am I right? Can your method deal with zero analysis error?"*

This could be done with considering the PKF applied on the dynamics of the model error as given by Eq.(4) in Nicolis (2003) – that is not done here. It could be interesting to investigate this problem from Eq.(19) with the PKF since the model error is specified with respect to the analysis state. While we do not discuss this point here, it can be a perspective of the work.

4. *"I am not sure to understand what is the predictability error # p . I guess it is just defined equation (14). So, in the linear approximation, # p is the propagation in time of the analysis error # a with the erroneous model M. Am I right? If not, it may alter my previous comments but then I would not understand the equation (33)."*

Yes, the predictability error, defined in Eq.(11), refers to the TL evolution of the analysis error by the numerical model M. But then, it can also apply for the nature, where the forecast error coincide with the predictability error. In order to clearly state the difference, we chose to add a \widetilde for the statistics when the dynamics is the nature (see p3, l22):
" In the latter, the predictability-error associated with the dynamics of the nature play an important role. So in order to avoid any confusion with the predictability error associated with the numerical model, the notation $\widetilde{\cdot}$ is used when the dynamics is the nature"

6. *"Several equations appeared separated in two, like in lines 110 and 264. Perhaps, this is just due to the draft template."*

*Yes, the separation in the equations is due to draft template only used for the peer review evaluation.*

5. *"line 227 : trom 7→ from"*

*The manuscript has been modified.*

5. *"line 255 : a space is missing at the end of the sentence"*

*The manuscript has been modified.*

[revised manuscript text omitted]

---

## Author Comment (AC2) · 14 Sep 2020

Final answer to the referee 2

First of all, we would like to thank the referee for her/his review on our paper and for giving us the opportunity to improve our paper.

Now, we organized the answer to the comments as follows. First, we list some changes afford to the manuscript then detail our answers to the questions raised by the referee.

**List of changes for the revision**

*Major changes*

The formalism has been reworded to match the usual notation of data assimilation (Sec. 2). In particular Sec. 2.1 has been split so to better introduce the modelling of the model-error covariance matrix that is now presented in Sec. 2.2. To this end, a new figure 1 has been introduced to sketch the dynamics of the uncertainty in presence of model error and to facilitate the setting of the framework considered here: the case where the numerical model is (more) dissipative compared to the nature. The formalism of the model error has been clarified, with the details of formulations of the forecast error that are given in Appendix A. A specific notation has been introduced for the predictability error of the nature ($\widetilde{\eps}^p$) so to avoid any confusion with the forecast error as usually encountered in data assimilation. Now, every $\widetilde{.}$ symbols refer to the uncertainty of the nature.

Sec 4.2 has been split in two part, one for the nature, the other for the model. The dynamics of the average over the model has been introduced to quantity the evolution of the variance and the length-scale fields. In particular, the Sec. 4.2.2 provides an analytical evolution of the predictability-error variance of the model which better explore the PKF equations. This quantitative evolution of the predictability-error variance is also now explored in section 4.3 for the model-error variance.

A section 5 has been introduced to discuss the results and the connexion with the previous work of Ménard et al 2000s that support the interest of the method introduced here, but in a real data assimilation system.

*Minor changes*

The detail of the ensemble estimation was missing in the previous version of the manuscript, this has been introduced in the introduction of Sec. 4.2.

There was an error on the definition of the time representation of Fig. 3 and later where the results are shown every $0.1T$, moreover the lead time of the simulation has been indicated $T=2.0$ (this was missing).

P10, line 12: The chordal distance was used in the numerical simulation, but we wrote the distance as |x-y| in the manuscript. While there is no difference between the two distances with the value of D and l_h used here, the chordal distance leads to a valid correlation function on the circle (that is not the case with the previous distance – this theoretical consideration is not discussed in the manuscript but justifications can be found in the article Pannekoucke et al. 2018).

*Differences between the two version of the manuscript*

To facilitate the comparison between the two version of the manuscript, a companion version of the manuscript lists all the modifications where old (new) statements are in red (blue).

**Answer to the question of the referee**

We copied your commentary in italics below, we reply in normal blue font.

*1. "Please discuss the condition that the assumption of the decorrelation between the analysis error and the model error is valid."*

Actually, this assumption may never apply in the real world. However, since it is hard to account in the modelling of the model-error it is often introduced with the consequence that the resulting modelling of the model-error certainly lead to over-estimate the "true" forecast-error statistics: the role of the cross covariance terms is to reduce the variance. This is now discussed in the new Sec. 2.2 (p3, l43-50):
"Then, assuming a decorrelation between the analysis and the model errors is certainly wrong for deterministic error as the model error due to the discretization of the dynamics; but it might apply for highly non-linear processes as for the turbulent processes and transport by the turbulent. Again, assuming the decorrelation between the analysis and the model errors leads to over-estimate the true effect of the model-error with an over-estimation of the true forecast-error uncertainty."

*2. "Since the prediction error is obtained from the tangent linear model, its validity depends on the error magnitude and the forecast range. Thus the method proposed in this paper also depends on the error magnitude and the forecast range? Please discuss."*

Thank you very much for the comment. We agree with this limitation and we have introduced a section dedicated for the discussion where this question is now addressed in the discussion part Sec. 5 among other points. In particular, we answered to the referee's comment as follows:

"While the PKF is designed from the TL approximation, it is a second order Gaussian filter that is a particular implementation of non-linear Kalman filter (Cohn, 1993):
for non-linear dynamics, the PKF equation of the mean state depends on the second order moments. However, for long-term predictions, or when the magnitude of the error is too large, the PKF would fails to provide an accurate estimation of the covariance matrices." (from p16, l 4)

*3. "There are many formulas in the paper, please make sure all the symbols have been explained. For example, what is the meaning of subscript "q"?"*

The subscript $q$ is for the time. This is now explicitly mentioned in the manuscript at p2 line 47. We have checked for other symbol to be sure that they were correctly explained.

*4. "To what extend can the PKF approximation provide the estimation of model-error covariance characteristics? Will it depend on the complexity of the model?"*

The PKF approximation can applies for non-linear dynamics by considering the TL dynamics of the uncertainty at the second order (see the answer to point 2). Actually, a closure appears even for linear dynamics when the order of the spatial derivative is larger than 1. Here, a closure is needed for the diffusion term that appears in Eq. (34). The closure we have used is the one introduced by Pannekoucke et al. (2018). As now better discussed in Sec. 5, the PKF is ready to be used for the tracer dynamics in 1D as well as in 2D and 3D domains. However, for multivariate dynamics, the PKF has to be developed. These limitations are now clearly stated in Sec. 5 (see p16, l31-43):

[revised manuscript text omitted]

---

## Author Comment (AC3) · 14 Sep 2020

Final answer to the referee 3

First of all, we would like to thank the referee for her/his review on our paper and for giving us the opportunity to improve our paper.

Now, we organized the answer to the comments as follows. First, we list some changes afford to the manuscript then detail our answers to the questions raised by the referee.

**List of changes for the revision**

*Major changes*

The formalism has been reworded to match the usual notation of data assimilation (Sec. 2). In particular Sec. 2.1 has been split so to better introduce the modelling of the model-error covariance matrix that is now presented in Sec. 2.2. To this end, a new figure 1 has been introduced to sketch the dynamics of the uncertainty in presence of model error and to facilitate the setting of the framework considered here: the case where the numerical model is (more) dissipative compared to the nature. The formalism of the model error has been clarified, with the details of formulations of the forecast error that are given in Appendix A. A specific notation has been introduced for the predictability error of the nature ($\widetilde{\eps}^p$) so to avoid any confusion with the forecast error as usually encountered in data assimilation. Now, every $\widetilde{.}$ symbols refer to the uncertainty of the nature.

Sec 4.2 has been split in two part, one for the nature, the other for the model. The dynamics of the average over the model has been introduced to quantity the evolution of the variance and the length-scale fields. In particular, the Sec. 4.2.2 provides an analytical evolution of the predictability-error variance of the model which better explore the PKF equations. This quantitative evolution of the predictability-error variance is also now explored in section 4.3 for the model-error variance.

A section 5 has been introduced to discuss the results and the connexion with the previous work of Ménard et al 2000s that support the interest of the method introduced here, but in a real data assimilation system.

*Minor changes*

The detail of the ensemble estimation was missing in the previous version of the manuscript, this has been introduced in the introduction of Sec. 4.2.

There was an error on the definition of the time representation of Fig. 3 and later where the results are shown every $0.1T$, moreover the lead time of the simulation has been indicated $T=2.0$ (this was missing).

P10, line 12: The chordal distance was used in the numerical simulation, but we wrote the distance as |x-y| in the manuscript. While there is no difference between the two distances with the value of D and l_h used here, the chordal distance leads to a valid correlation function on the circle (that is not the case with the previous distance – this theoretical consideration is not discussed in the manuscript but justifications can be found in the article Pannekoucke et al. 2018).

*Differences between the two version of the manuscript*

To facilitate the comparison between the two version of the manuscript, a companion version of the manuscript lists all the modifications where old (new) statements are in red (blue).

**Answer to the question of the referee**

We copied your commentary in italics below, we reply in normal blue font.

*1. "Equations throughout the discussion version are poorly formatted and awkwardly split, this should be fixed for better readability. This could also use hyperlinking in equation references to make the document more easily searchable."*

We send the new version of the manuscript as two-column which is better presented.

*2. "I agree with the other review comments that the definitions of quantities such as the model error $\eps^m_{q+1}$, and their evolution equations, need to be more clear."*

The new version of the manuscript clarify the presentation of the data assimilation so to avoid any confusion in Sec 2.1. Moreover, the introduction of the modelling of the flow-dependent of the model-error covariance matrix is now clarified in the new section 2.2.

Moreover, the new appendix A provides the derivation of the decomposition of the forecast error which contributes to clarify the manuscript.

*3. "My understanding is that the CFL condition is stated as, $C = u\delta t\delta x \leq C\ max$ where $C\ max$ is typically taken equal to 1. The CFL condition is stated differently in lines 219, page 8, and 301, page 11, and it does not seem to me that these statements correspond to the above condition. Please check these lines for consistency."*

Sorry this was a typos. It is now corrected. Thank you.

*4. "In appendix B, lines 495 - 499, page 21: when going through the derivation of the modified equation for the Euler scheme, I find that $\kappa = u /2 (\delta x + u\delta t )$ as the term $-u^2 \delta t /2\ 2\ \partial x\ \tilde{c}$ appears on the left-hand-side of the equation on line 495. Please verify the equations for consistency and update the discussion in lines 238 - 246, page 9. Does this affect any of the numerical results such as Fig 1.b? Are there any other places where this would change the interpretation of the results?"*

We would like to thank the referee for his/her comment, actually this was a typos in the retranscription of the derivation given in Appendix B, where a minus in the line 491 has been replaced by a plus in the line 492 (of the previous version of the manuscript). And this error has been propagated in the manuscript in line 495. This is now corrected in the appendix B of the new manuscript. With this correction, you will find that the diffusion coefficient $kappa = u/2(dx -u dt)$. Because it was a typos and not a true error in the manuscript, and that all the simulation has been computed with the correct expression of kappa, nothing has been changed elsewhere.

*5. "The conclusion of the numerics is too short. The main point of the work, estimating the model error covariance via the PKF formalism is somewhat successful, and the limitation of the decorrelated error assumption is discussed. However, there isn't much quantitative analysis of practical use of this model error estimate. For instance, can the estimated variance in the PKF formalism be used as an upper bound for the true model error variance?"*

New quantitative elements have been introduced in Sec 4.2.2 and Sec. 4.3 in order to feature the characteristic of the part of the model error that is introduced here.

In the new version of the conclusion, the quantitative impact of the flow-dependent modelling, explicitly proposed here, relies on the results of Ménard 2000s who measured a positive impact of using this approach in the assimilation of stratospheric tracer. This is discussed in the new section 5. As the flow-dependent modelling Eq.(18) is introduced when the numerical model is dissipative while the nature is not -- or when the numerical model is more dissipative than the nature –, the estimated variance in the PKF should be considered as a lower bound of the true model-error variance, that should be completed by the residual variance Q. This is now stated in the conclusion (see p17, l13-18):

"The flow-dependent formulation being introduced for modelling the situation where the numerical model is more dissipative than the nature, the model-error variance provided by the PKF should be a lower bound of the true model-error variance, that need a residual climatological covariance to account for the bias. "

6. "Page 5, lines 121 - 123: I find the parentheses in this sentence to be awkward, as it is easy to confuse their use as mathematical arguments. The conditional statements for the forecast, respectively the analysis, should use commas to clarify the meaning of the text."

This has been rephrased. Than you.

7. "Page 10, line 255: see the typo "details.Note"."

This has been corrected, thank you.

8. "Page 13, lines 343 - 344: the definition of N sentence doesn't seem to be complete. I find the notations confusing here, where N is referred to as a model equation. In this context, what equations represent M?"

The N / M notation in Sec. 4.1 have been verified. There were some typos. Thank you.

9. "Page 15, line 385: see typo "legnth-scale"."

This has been corrected, thank you.

10. "Page 21, Eq. C1: see typo $x\,k_{i-1}$ ."

This has been corrected, thank you.

[revised manuscript text omitted]

---

## Referee Report (RR1)

The authors have achieved a great job with this revised version. The presentation, the notations, the model assumption (typically approximation (27): neglecting Q in  $P^m$  computation), the mathematical and intuitive arguments which support this model are much clearer. There are no more ambiguities. The authors have answered all the questions of my review report. In particular, I thank the author for the reference to the work of Nicolis, which seems particularly relevant.

Yet, since I now well understood the main assumptions of the draft model, it raises other questions (or rather, it makes me reformulate some of my previous questions).

 Mainly, I wonder what is the validity of the approximation (27). Actually, it may be a more severe assumption than assuming the decorrelation between model error and analysis error. I understand that all your methodology is built on top of this assumption (27); that relaxing this assumption has to be left for future works. And in my opinion, even if approximation (27) is debatable, this should not prevent your work to be publishable.

But, at least, you could check, in your numerical test case, the order of magnitude of the 3 terms of equation (21):  $P^m$  (the "true" one, ie not the one approximated from (27)),  $\Pi^m$  and Q. You may focus on the matrix trace (mean of variances). For instance, problems could appear if  $P^m \sim Q \gg \Pi^m$  (when e.g. the already-discussed case  $\epsilon^a \ll 1$ ) or if  $P^m \ll \Pi^m \sim Q$ . Indeed,  $\Pi^m$  could be negative (even

though it is probably positive in your diffusive example) in some cases and may balance Q in (27).

At page 15 (around line 45), orders of magnitude are compared, but it seems to me that no value is discussed for the "true" model error variance.

2) I do not understand the terminology "flow-dependent part of  $P^{m}$ " for  $\Pi^m$  and "climatological part /bias of  $P^{m}$ " for Q. It seems to me that both terms are flowdependent, isn't it? Is the bias terminology come from the type of  $\epsilon_{q+1}^m(\chi^a)$ expression which depends on  $\chi^t$ ?  $\left(\epsilon_{q+1}^m(\chi^a) \approx \int_{t_q}^{t_{q+1}} [(v - U) \cdot \nabla - \kappa \Delta] \chi^a dt\right)$ But then, the same description/ terminology could be used for  $\epsilon_{q+1}^m(\chi^t)$  and thus  $P^m$ because the  $\epsilon_{q+1}^m(\chi^t)$  expression is similar. And  $P^m$  is the initial quantity of interest.

**Typo and small corrections:**

- $\Rightarrow$  Fig 1 : the « red » looks more like a salmon
- $\Rightarrow$  ^m is a the wrong place in equation 6
- ⇒ Page 5, line 23, (2.1) => (21)
- $\Rightarrow$  Equation 47a, V => V^p

---

## Referee Report (RR2)

**1 General response**

I want to thank the authors for the work that they put into the revision, refining various aspects of the derivation of the forecast error covariance, more clearly distinguishing the predictability error and its connection to the state-dependent model error source. I think these additional derivations have addressed the earlier questions and have added strength to the overall analysis in the work. Likewise, I thank the authors for addressing the typos and minor errors found in the last version. Generally I am quite positive about the work and I would only like to make the following minor and technical suggestions. Following these minor points I recommend publication.

**2 Minor Points**

1. Page 1, lines 39 - 41: I think there are a few other studies worth mentioning that explicitly link numerical discretization error (or lack of numerical precision in a general sense) to the data assimilation cycle and model bias. Though these do not explicitly discuss estimation the model error covariance as in the present study, these works are worth mentioning in this general context: Dubinkina [2018], Hatfield et al. [2018], Grudzien et al. [2020].

2. Page 3, lines 40 - 41: "For instance, assuming that the model error is unbiased, leads to model the bias". I believe this should be rephrased as "leads to modeling the bias".

3. Page 3, lines 47 - 50: "Again, assuming the decorrelation between the analysis and the model errors leads to over-estimate the true effect of the model-error with an over-estimation of the true forecast-error uncertainty." I believe this should be re-phrased as "leads to over-estimating the true effect".

4. Page 3, lines 91 - 92 : "in that case, a model-error is needed". I believe this should should read "in that case, a model-error estimate is needed".

5. Page 3, line 95: "larger enough" should read "large enough".

6. Page 4, line 41: "not necessary"' should read "not necessarily".

7. Page 5, lines 16 - 18: grammar should be checked on this sentence and clarified.

8. Page 5, line 77: "(assume)" should read "(assumed)".

9. Page 6, Eq. (23): bracket is misaligned in the $\mathbf{g}$ norm.

10. Page 12, second column, first line: see typo "ad".

11. Page 18, appendix A2: The equation between (A4) and (A5) has a misplaced transpose in the equation,

$$\left(\mathbf{N}\mathbf{P}^a\mathbf{M}^T\right)^T \tag{1}$$

**References**

S. Dubinkina. Relevance of conservative numerical schemes for an ensemble kalman filter. *Quarterly Journal of the Royal Meteorological Society*, 144(711):468–477, 2018.

C. Grudzien, M. Bocquet, and A. Carrassi. On the numerical integration of the lorenz-96 model, with scalar additive noise, for benchmark twin experiments. *Geoscientific Model Development*, 13(4):1903–1924, 2020.

S. Hatfield, P. Düben, M. Chantry, K. Kondo, T. Miyoshi, and T. N. Palmer. Choosing the optimal numerical precision for data assimilation in the presence of model error. *Journal of Advances in Modeling Earth Systems*, 10(9):2177–2191, 2018.

---

## Author Response (AR2)

Final answer to the referee 1 (report #2)

First of all, we would like to thank the referee for her/his review on our paper and for giving us the opportunity to improve our paper.

**Differences between the two version of the manuscript**

To facilitate the comparison between the two version of the manuscript, a companion version of the manuscript lists all the modifications where old (new) statements are in red (blue).

Note that a reference for Eq.(25) where missing we add one.

**Answer to the question of the referee**

We copied your commentary in italics below, we reply in normal blue font.

*1.a) "Mainly, I wonder what is the validity of the approximation (27). Actually, it may be a more severe assumption than assuming the decorrelation between model error and analysis error. I understand that all your methodology is built on top of this assumption (27); that relaxing this assumption has to be left for future works. And in my opinion, even if approximation (27) is debatable, this should not prevent your work to be publishable."*

We agree that this needs further study.

*1.b) "But, at least, you could check, in your numerical test case, the order of magnitude of the 3 terms of equation (21) : P m (the 'true' one, ie not the one approximated from (27)), Π m and Q. You may focus on the matrix trace (mean of variances). For instance, problems could appear if P m ~Q ≫ Π m (when e.g. the already-discussed case ε a ≪ 1) or if P m ≪ Π m ~Q. Indeed, Π m could be negative (even though it is probably positive in your diffusive example) in some cases and may balance Q in (27)."*

Well, the problem is that we do not know what is the true model error covariance $P^m$ – as mentioned at the end of section 4.3 we need to estimate Q, that can not be conducted here (need long time experiment). But this could be done by considering a dynamics for which conducting long term integration and assimilation experiment is possible (eg Kuramoto-Shivashinsky). This could be done in a future work.

*1.c) "At page 15 (around line 45), orders of magnitude are compared, but it seems to me that no value is discussed for the 'true' model error variance."*

As mentioned in the answer to the previous part of the question (see the above 1.b), the true model-error variance is not known (while the model-error can be computed!) because we need to compute the static covariance matrix Q, so here we can not compare with the true model-error variance.

*2. "I do not understand the terminology 'flow-dependent part of $P^m$' for $\Pi^m$ and 'climatological part /bias of $P^m$' for Q. It seems to me that both terms are flow-dependent, isn't it? Is the bias terminology come from the type of $\varepsilon_{q+1}(\chi_a)$ expression which depends on $\chi_t$ ? ($\varepsilon_{q+1} \approx \int t_{q+1} [(v - U) \cdot \nabla - \kappa\Delta] \chi_a \, dt$) But then, the same description/ terminology could be used for $\varepsilon^m_{q+1}(\chi_t)$ and thus $P^m$ because the $\varepsilon^m_{q+1}(\chi_t)$ expression is similar. And $P^m$ is the initial quantity of interest."*

The terminology refers to the decomposition of the model-error covariance matrix $P^m$, not to the

decomposition of the error. In particular, we agree that the errors related to the flow-dependent/static part of $P^m$ are both flow-dependent. However, the Q matrix, resulting of the climatological effect of the bias in the model-error decomposition, is static (not flow-dependent, while the value depends on the duration of the integration). So the terminology concerns the matrix and that should not be confused with the error itself.

We precise this terminology in the manuscript by adding the following comment (p4, l14-26):

"To account for the bias, a climatological residual covariance matrix would be necessary, that corresponds to a static matrix which depends to the duration of the forecast. Note that this decomposition of the model-error covariance matrix into a flow-dependent and a static part, should not be confused with the decomposition of the model error itself. In particular, the decomposition of the model-error covariance matrix does not mean that the part of the model error related to the bias is static, this not true, the bias depends on the situation. However, the estimation of the bias need the knowledge of the nature dynamics that is never known. Because the statistical contribution of the bias can only be known from a climatological study, this leads to a static matrix which is not flow-dependent."

Where the underlined part correspond to what has been added in the manuscript.

*3. Typo and small corrections:*

*3.1) Fig 1 : the « red » looks more like a salmon*

The colour is red, but the transparency used may change the aspect depending on the medium (screen or paper). We preferred to keep the "red" colour name in the manuscript.

*3.2) $^m$ is a the wrong place in equation 6*

This has been corrected, thank you.

*3.3) Page 5, line 23, (2.1) => (21)*

Sorry, we did not find the modification suggested by the referee.

*3.4) Equation 47a, V => $V^p$*

This has been corrected, thank you.

First of all, we would like to thank the referee for her/his review on our paper and for giving us the opportunity to improve our paper.

**Differences between the two version of the manuscript**

To facilitate the comparison between the two version of the manuscript, a companion version of the manuscript lists all the modifications where old (new) statements are in red (blue).

Note that a reference for Eq.(25) where missing we add one.

**Answer to the technical corrections mentioned by the referee**

We copied your commentary in italics below, we reply in normal blue font.

*1. "Page 1, lines 39 - 41: I think there are a few other studies worth mentioning that explicitly link numerical discretization error (or lack of numerical precision in a general sense) to the data assimilation cycle and model bias. Though these do not explicitly discuss estimation the model error covariance as in the present study, these works are worth mentioning in this general context: Dubinkina [2018], Hatfield et al. [2018], Grudzien et al. [2020]."*

The references have been added in the revised manuscript.

*2. "Page 3, lines 40 - 41: "For instance, assuming that the model error is unbiased, leads to model the bias". I believe this should be rephrased as "leads to modeling the bias"."*

This has been rephrased, thank you.

*3. "Page 3, lines 47 - 50: "Again, assuming the decorrelation between the analysis and the model errors leads to over-estimate the true effect of the model-error with an over-estimation of the true forecast-error uncertainty." I believe this should be re-phrased as "leads to over-estimating the true effect"."*

This has been rephrased, thank you.

*4. "Page 3, lines 91 - 92 : "in that case, a model-error is needed". I believe this should should read "in that case, a model-error estimate is needed"."*

This has been modified, thank you.

*5. "Page 3, line 95: "larger enough" should read "large enough"."*

This has been corrected, thank you.

*6. Page 4, line 41: "not necessary"' should read "not necessarily".*

This has been corrected (also modified in P5l6), thank you.

*7. Page 5, lines 16 - 18: grammar should be checked on this sentence and clarified.*

This has been corrected, thank you.

*8. Page 5, line 77: "(assume)" should read "(assumed)".*

This has been corrected, thank you.

*9. Page 6, Eq. (23): bracket is misaligned in the g norm.*

This has been corrected, thank you.

*10. Page 12, second column, first line: see typo "ad".*

This has been corrected, thank you.

*11. Page 18, appendix A2: The equation between (A4) and (A5) has a misplaced transpose in the equation, $(NP^a M^T)^T$*

We permute N and M. This has been corrected, thank you.

[revised manuscript text omitted]